# VLASim: World Modelling via VLM-Directed Abstraction and Simulation

## Abstract

Generative video models, a leading approach to world modeling, face fundamental limitations. They often violate physical and logical rules, lack interactivity, and operate as opaque black boxes ill-suited for building structured, queryable worlds. To overcome these challenges, we propose a new paradigm focused on distilling an image caption pair into a tractable, abstract representation optimized for simulation. We introduce VLASim, a framework where a Vision-Language Model (VLM) acts as an intelligent agent to orchestrate this process. The VLM autonomously constructs a grounded (2D or 3D) scene representation by selecting from a suite of vision tools, and accordingly chooses a compatible physics simulator (e.g., rigid body, fluid) to act upon it. VLASim can then infer latent dynamics from the static scene to predict plausible future states. Our experiments show that this combination of intelligent abstraction and adaptive simulation results in a versatile world model capable of producing high quality simulations across a wide range of dynamic scenarios.

## 1 Introduction

Understanding and forecasting how the visual world evolves is a core challenge for building intelligent systems. Humans can observe a static scene and infer not only its current structure but also how it might change over time and in response to different actions. This capability underlies essential skills such as planning, decision-making, and causal reasoning. Replicating this physical intuition in AI hinges on creating robust "world models" to predict potential futures. In recent years, the dominant paradigm for building such models has been large-scale video generation. By training on immense visual corpora, these models have achieved remarkable success in synthesizing complex, dynamic scenes with impressive visual realism, suggesting they are learning a powerful, albeit implicit, model of our world only from 2D observations and text descriptions (Wan et al., 2025; Google Deepmind, 2025b). However, as most video models are only conditioned on image and text inputs, they lack a native mechanism for physical interaction, which limits their utility for tasks that require physical intuition. While some models incorporate action conditioning (Song et al., 2025; Google Deepmind, 2025a; Wang et al., 2023), the action space is typically limited to simple transformations, such as changes in camera viewpoint, rather than complex physical interventions. Where models do incorporate physical interventions, these are typically only for application in narrow domains such as robotic manipulation (Zhu et al., 2024) or driving (Wang et al., 2023).

Despite their visual prowess, pixel-space models result in critical and systematic failures that limit their use for robust interaction (Motamed et al., 2025; Kang et al., 2024). First, they frequently generate physically implausible scenarios. Video models learn statistical correlations from pixels and do not enforce any physical plausibility constraints, which results in their outputs often violating fundamental principles of object permanence, collision, and causality. For example, the number and size of objects can change erratically, or objects can accelerate without a corresponding force. This failure to deduce underlying principles is not limited to 3D physics; these models are equally unable to infer the structured representations and simple, deterministic rules required to simulate abstract 2D environments like Conway's Game of Life (Conway et al., 1970). Second, these models operate as opaque 'black boxes'. The generated scene is not a structured, queryable world but a sequence of pixels. Consequently, it is impossible to inspect the environment's underlying state or apply novel physical actions beyond those observed during training. Ultimately, the world inside these models is a passive movie to be watched, not a dynamic environment to be acted upon. To build

useful world models, we need approaches that yield structured, interactive, and physically grounded representations.

Separate from generative video models, another significant line of research has focused on reconstructing 3D or 4D scene representations from images. Foundational models like DUST3R (Wang et al., 2024) have demonstrated impressive capabilities in producing dense and accurate geometric reconstructions, while methods based on Neural Radiance Fields (NeRFs) (Mildenhall et al., 2021) excel at generating photorealistic novel views. However, the primary objective of both lines of work is to capture a scene's geometry and appearance, not its underlying physical nature. This focus on 3D geometry also renders them, by design, inapplicable to abstract 2D environments governed by logical rules, such as cellular automata like Conway's Game of Life and grid-based games like Snake. The resulting representations, e.g., point clouds or radiance fields, are not amenable to tractable simulation. Most methods do not decompose scenes into simplified primitives or, crucially, infer the physical properties (e.g., mass, friction, elasticity) necessary for a physics engine. Consequently, while these methods can show what a scene looks like from a new angle, they cannot predict what will happen next in response to physical forces, leaving a critical gap for simulation-ready world models. While some follow-up work has attempted to retrofit these representations for 3D physics (Zhang et al., 2024b; Petitjean et al., 2023), such efforts are typically restricted to narrow classes of simulation and fall short of a truly versatile and generalizable system.

In this paper, we introduce `VLASim`, a novel framework for world modeling that moves away from direct pixel prediction and instead builds an explicit, structured world representation. Instead of predicting pixels, our primary goal is to distill a visually complex image into a tractable abstract representation, illustrated in Figure 1. This representation intentionally discards physically irrelevant information (like fine-grained textures or static backgrounds) to cre-

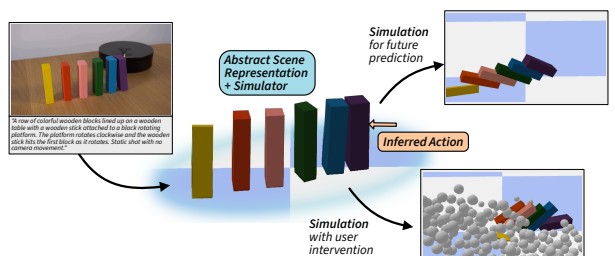

Figure 1: Overview of `VLASim`.

ate a structured world model optimized for simulation. Our framework achieves this through a Vision-Language Model (VLM) that acts as a central agent, orchestrating three core innovations, also refer to Figure 2. First, the VLM acts as an intelligent tool-using agent to construct a grounded representation. It is equipped with a versatile suite of vision modules, including segmentation, 3D reconstruction, and primitive fitting, and autonomously decides which tools to deploy. This allows the representation to be grounded in the scene's native dimensionality; for instance, applying the full 3D pipeline for spatial environments while recognizing that such tools are irrelevant for planar ones, such as Conway's Game of Life. Second, the choice of representation and simulator is co-dependent and adaptive. The VLM jointly determines the type of abstraction and the most appropriate physics simulator to act upon it: a scene with blocks is abstracted into a rigid body model and paired with a rigid body solver, while one with water is represented as a particle system paired with a fluid dynamics engine. Finally, this structured world model enables `VLASim` to infer latent dynamics, predicting a scene's likely evolution from the image and caption alone based on visual cues and the description of the scene. Through comprehensive experiments, we show this combination of intelligent abstraction, adaptive simulation, and inferred dynamics results in a world model that significantly outperforms prior methods in producing high-quality, physically and logically plausible simulations across a wide range of scenarios.

## 2 Previous Work

**Video Models**  Recent advances in generative methods have established large-scale video models as a dominant paradigm for modeling world dynamics. State-of-the-art models have demonstrated a remarkable ability to synthesize high-fidelity and temporally coherent videos from text and image inputs (OpenAI, 2024; Google Deepmind, 2025b; Runway, 2025; Wan et al., 2025). These models use diffusion or flow-matching approaches in a compressed latent space for video generation. While most models only condition on image and text inputs, some recent methods have enabled

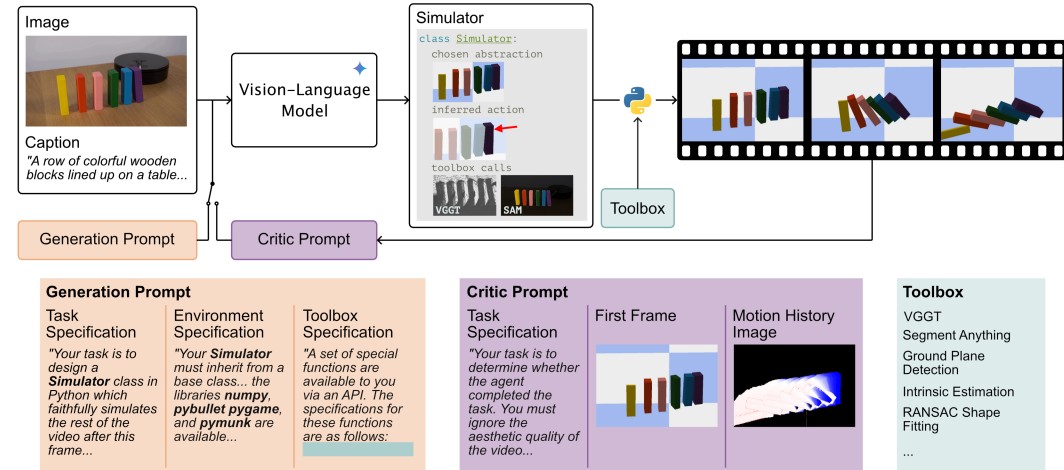

Figure 2: Illustration of our method. VLASim takes a single image and text caption as input. We design a generation prompt that the VLM uses to generate a simulator of the scene, including the scene abstraction, actions that can be inferred from the input, and calls to the toolbox. The simulator code is then executed to generate future predictions. We can also generate other diverse videos by interactively changing the actions. Finally, a code critic step is used to automatically make changes to the Simulator to correct any errors and improve the quality of the simulation.

conditioned on other parameters, such as camera parameters, for more controllable video generation (Huang et al., 2025; Song et al., 2025; Google Deepmind, 2025a). Additionally, some models demonstrate conditioning on action inputs, but these tend to be limited to action in a controlled domain, such as robotic manipulation tasks (Zhu et al., 2024) or driving (Wang et al., 2023). However, it is very difficult to interact with these models outside of the control parameters used during training. It is generally impossible to query the state of an object, apply a novel physical force, or explore alternative outcomes under different conditions. These models do not explicitly reason in a structured space, and instead directly perform computations in frame space. This leads their outputs to frequently violate fundamental principles of the real world (Motamed et al., 2025; Li et al., 2025; Kang et al., 2024). Common failure cases include the violation of object permanence, where objects may inexplicably appear or vanish, and inconsistent causality, where actions do not have plausible consequences. The lack of explicit, structured reasoning and interactivity in pixel-space video models limits their utility as robust world models, motivating our shift towards an explicit, simulation-based approach.

**Scene Reconstruction and Simulation**    3D reconstruction from images has achieved considerable success in recent years. Models like DUST3R (Wang et al., 2024) and its follow-ups (Zhang et al., 2024a; Wang et al., 2025; Feng* et al., 2025) produce dense geometric reconstructions from images, while methods based on Neural Radiance Fields (NeRFs) (Yu et al., 2021; Tewari et al., 2023) and 3D Gaussians (3DGS) (Szymanowicz et al., 2024; Charatan et al., 2024) and their 4D extensions (Wu et al., 2024; Tretschk et al., 2021; Yang et al., 2023; Yunus et al., 2024) excel at generating photorealistic novel views. While most approaches do not inherently decompose the scene into discrete, object-centric components, some methods have made progress with the help of features from vision-language models (Kerr et al., 2023; Jatavallabhula et al., 2023). Additionally, most neural radiance methods are optimized for appearance rather than physics, making them poorly suited for interactive simulation. Some work has attempted to model physics, for instance, by combining 3D representations with physics engines (Zhang et al., 2024b; Petitjean et al., 2023; Feng et al., 2024; Li et al., 2023; Wu et al., 2015; Le et al., 2025; Chen et al., 2025; Kairanda et al., 2025; Xie et al., 2024), but they all only model limited physical phenomena. Concurrent work, PhysGen3D (Chen et al., 2025), reconstructs object-centric 3D scenes and performs simulation with a fixed material point method. In addition, it only takes an image as input and cannot reason about input text. In this work, we use tools from scene reconstruction and simulation methods but do not use a fixed pipeline as the VLM is free to select scene representations and simulators best suitable for any input.

**Program Synthesis with VLMs**   Our work is informed by recent advances in using Vision-Language Models (VLMs) as agents that synthesize programs to solve complex visual tasks. A foundational paradigm is visual program synthesis for querying. Methods like ViperGPT (Surís et al., 2023) and VisProg (Gupta & Kembhavi, 2023) can parse a complex visual query into a sequence of steps, generating code that calls various vision APIs (e.g., object detectors, depth estimators) to arrive at a final answer. LayoutGPT (Feng et al., 2023), which uses an LLM to generate a complete scene layout, including the sizes, positions, and relationships of different objects. Other research has shown that a VLM can evolve interpretable visual classifiers (Chiquier et al., 2024) and design interpretable programs to describe underlying scientific laws (Mall et al., 2025). A second major application is in high-level planning and robotics. This research aims to create agents that can reason about the world to perform actions. VisualPredicator (Liang et al., 2024), for example, learns neuro-symbolic predicates that classify the state of the world for a symbolic planner. Others, like VoxPoser (Huang et al., 2023), use LLMs to synthesize 3D affordances that guide a low-level motion planner. The common thread in this research is that the VLM's role is to generate a plan or a set of actions for an agent to execute within an existing environment.

## 3 VLASim: World Modelling via VLM-Directed Abstraction and Simulation

The input to `VLASim` is a pair consisting of a single image and a text prompt that describes the scene. The goal of `VLASim` is to convert this static input into a dynamic, interactive world model. This process is orchestrated by a central Vision-Language Model (VLM), which generates a complete "world program" in Python ready for execution. This program consists of three key components: (1) *A Grounded Abstract Representation*: The VLM selects from a suite of vision tools to construct a 2D or 3D model of the scene, optimized for simulation, (2) *Inferred Latent Dynamics*: It predicts the most likely implicit action from the visual and textual cues, which serves as the initial condition for the simulation, (3) *A Selected Simulator*: It determines the most compatible simulation engine (e.g., rigid body, fluid, logic) to simulate the scene's dynamics. Once generated, this world program is executed to predict a plausible future. Because the program describes an explicit and structured world, it can also be modified with novel user-defined interventions to imagine diverse futures.

### 3.1 Prompting for World Program Generation

The core of `VLASim` lies in guiding a powerful Vision-Language Model (VLM) to generate a complete, executable world program. Instead of fine-tuning, we steer the model's behavior at inference time using a comprehensive, multi-part prompt. The prompt begins with a high-level task specification in natural language. This instruction outlines the overall objective: to analyze a user-provided image and text description and produce a self-contained Python script that simulates the scene's future. This main directive then embeds two more structured components to formalize the task: environment specification that provides the structural code template, and toolbox specification, which provides the API definitions for the suite of perception tools the VLM can use.

**Task Specification**   The task specification is a high-level, natural language instruction. It directs the VLM to analyze the user-provided inputs and produce a self-contained Python script that simulates the scene's future, making use of the other prompts to structure its output and call the necessary tools. A condensed version is shown in Figure 3.

**Environment Specification**   The environment specification provides the formal scaffolding for the VLM's code generation task. Its central element is a Python Simulator base class that the VLM must use to derive the model from. This base class defines the core methods the VLM must implement, enforcing the entire simulation logic from scene setup to frame-by-frame execution. Additionally, the environment provides the VLM with a list of existing python libraries that it can rely on, pointing it to common simulation implementations. This ensures the VLM's output is structurally compatible with our execution environment, as illustrated in Figure 5.

**Tool Specification**   Finally, the tool specification provides the VLM with the API of a diverse toolkit used for scene understanding and simulation setup. These helper functions are not required

```
# ROLE: Computer Scientist

# TASK: Generate an executable Python class `VideoSimulation`
# that simulates the future of the scene from the input image
# and caption: "[CAPTION]".

# KEY PRINCIPLES:
# 1. Minimal Abstraction: Determine if the scene is fundamentally
# 2D or 3D and use the simplest required representation
# 2. Activating Agents: Model the *effect* of an external agent
# (e.g., a hand pushing a block), not the agent itself.
# 3. Robustness: Prioritize robustness to sensor noise.
```

Figure 3: A condensed version of the task specification provided to the VLM, outlining its role and key principles.

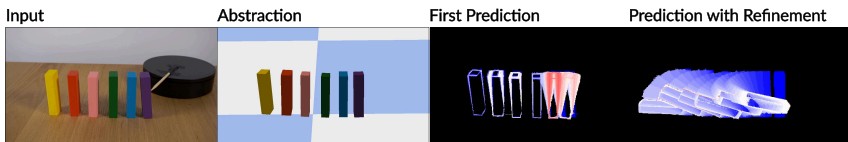

Figure 4: The VLM also acts as a critic. If the result is not perceived to be correct, a new result is generated. The spatiotemporal visualisations in the last two columns visualise the motion in the video as a static frame. In this example, the second block from the left has incorrect motion in the first prediction, that is then resolved with a better dynamics prediction in the last column.

```
class Simulator(Iterator):
    def __init__(self, frame_size=(1024, 576), api: API=None, fps=30):
        """Initializes the simulator."""
    def fit(self, image: np.ndarray, text: str):
        """Fit the simulator's parameters to the provided image."""
    def update_simulation(self, dt: float):
        """Update the simulation by one timestep dt."""
    def render_frame(self):
        """Render the next frame of the simulation."""
# Available libraries: numpy, scipy, pybullet, pygame, pymunk..
```

Figure 5: The environment specification implements the base class, which defines the required structure for the VLM's generated Python code, and also provides a list of relevant Python libraries.

to be used by the VLM, but often help in optimising the solution. The API is organized into three categories: (1) Core Perception tools for open-vocabulary segmentation and 3D point cloud estimation; (2) Geometric Processing functions for fitting planes, cleaning data, and abstracting objects into primitive shapes; and (3) Simulation Interface methods that directly add objects (e.g., rigid meshes, soft bodies, or particles) into physics engines. This rich set of tools allows the VLM to translate its conceptual understanding of a scene into the precise, low-level code required to instantiate and run a simulation, as exemplified in Figure 6.

## 3.2 PERCEPTION TOOLBOX

To power the API exposed in the tool specification, we implement a suite of perception and geometry modules that the VLM can call to perform a wide range of tasks. These functions are implemented to aid the inference of VLM. The VLM does not re-implement or update the implementation of these updates, and it is free to ignore these implementations if it does not find any use for them.

```python
class API:
    def segment(self, image: np.ndarray, objects: List[str]):
        """Segments the image."""
    def fit_3d_shape(self, point_cloud: np.ndarray, shape_class: str):
        """Fits a 3D primitive ('cuboid', 'sphere', etc.) to a
        point cloud and returns its parameters."""
    def generate_surface_mesh(self, vertices, indices, mass=0.0):
        """Creates a mesh from a vertex mesh."""
```

Figure 6: The API provided in the tool specification defines a rich set of functions for perception and geometric processing.

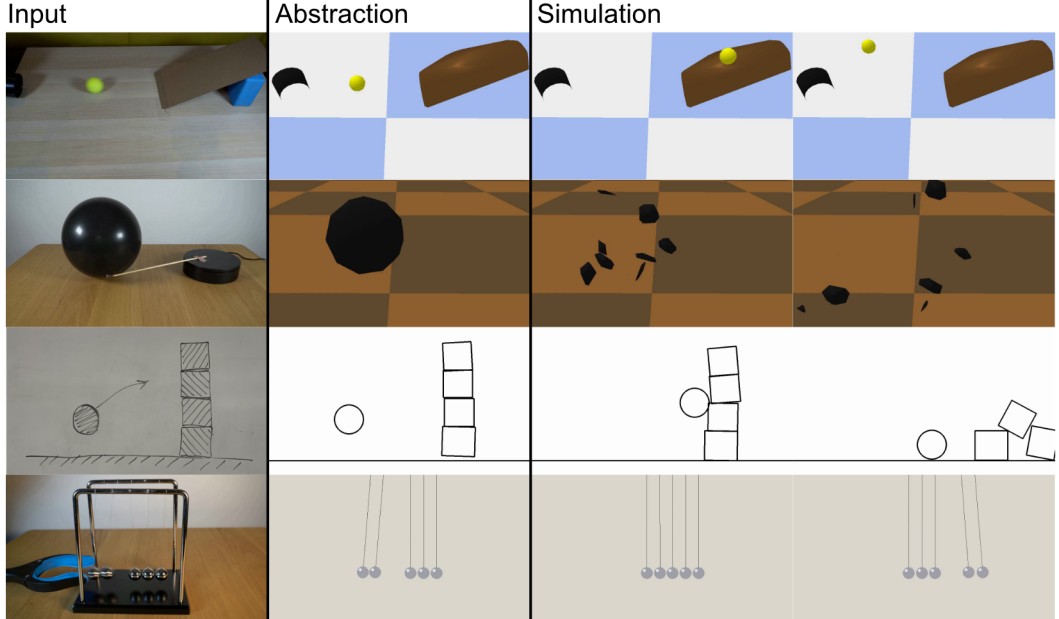

Figure 7: `VLASim` generates an abstraction of the scene and a simulator that can together be used to predict future scene states. From top to bottom: (Rigid Body) ball moving on an incline, (Thermodynamics) balloon bursting on interaction with flame, (Abstraction) A collision drawn on a whiteboard, and (2D Simulation) Newton's cradle.

### 3.2.1 2D PERCEPTION AND GEOMETRY

**Open-Vocabulary segmentation** The `segment` API function is built upon a state-of-the-art open-vocabulary segmentation pipeline. We first use Gemini Perception (Comanici et al., 2025) to estimate the bounding boxes for the query, and then Segment Anything (Kirillov et al., 2023) to compute dense segmentation maps from those boxes. The VLM infers the important and relevant objects in the scene to query this function. This function is used in almost all scenes to develop the right abstract scene representation.

**Geometry Helpers** The `fit_2D_shape` function fits simple 2D geometric primitives, such as disks and polygons, to selected regions. This helps in developing the right abstractions that can be used for tractable simulation.

### 3.2.2 3D PERCEPTION AND GEOMETRY

**Single Image 3D Estimation** We use VGGT (Wang et al., 2025) to estimate dense 3D point maps from a single image as the `pts3d` function. This model is also used to implement the `intrinsics` call that computes the camera parameters.

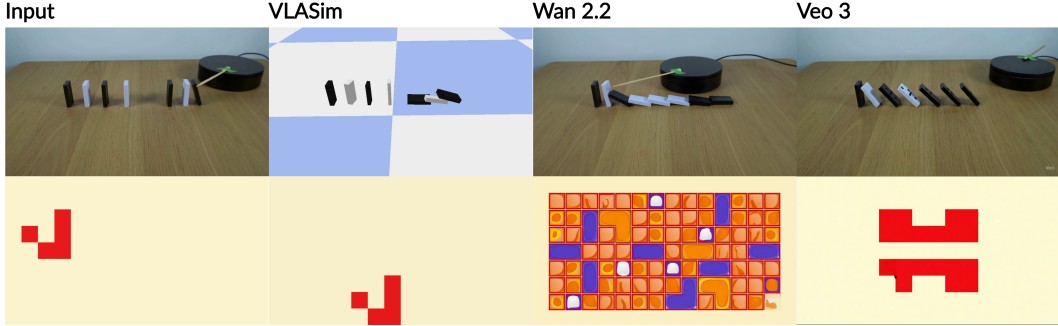

Figure 8: Comparisons of our approach with video generation models Wan 2.2 and Veo 3. Compared to both of these models, our approach is able to follow physical principles such as object permanence and the effect of gravity (Top); while on the Game of Life dataset our approach is able to correctly infer future patterns based on the true rules of the game (Bottom).

**Geometry Helpers**  These tools rely on established computer graphics algorithms. The `predict_ground_plane` function uses a RANSAC-based approach to robustly fit a plane to a point cloud, allowing the system to establish a world coordinate frame. The `fit_3d_shape` function performs robust, RANSAC-based, fitting of simple geometric primitives to point clouds. This is an important component, as it not only abstracts the shape for tractable simulation, but also computes a complete shape from incomplete point clouds. The `generate_surface_mesh` and `add_soft_body` create meshes and soft bodies from point clouds.

### 3.3 Critic and Code Refinement

While a single generation pass can produce high-quality results, high complexity in scenes can lead to errors in the initial code or inaccurate scene fits. To enhance the robustness of our system, we introduce an automated feedback loop: a two-stage Critic and Refinement process, as illustrated in Figure 2. This allows the system to identify and correct its own mistakes.

**Critic Stage.**  In the first stage, a VLM is prompted to act as a critic. Along with the text caption, the critic is provided with the initial frame of the generated simulation and, crucially, a spatiotemporal colormap that visualizes all dynamic activity over the simulation's duration (blue for early motion, red for late), see Figure 4 for a visualization. The critic's task is to assess the correctness of the simulation's initial conditions and physical setup, not its visual quality. It then outputs a structured JSON object containing a boolean flag evaluating the accuracy of the simulation, and a list of suggested improvements.

**Refinement Stage.**  If the critic deems the simulation inaccurate, the second stage begins. A VLM is prompted to act as a 'code refiner'. It receives the original, flawed Python code generated in the first pass, along with the specific suggested improvements from the critic's JSON feedback. Its task is to debug and rewrite the code to address the identified issues, producing a final, corrected `VideoSimulation` class. This self-correction capability improves the quality and physical plausibility of the final output.

**Automated Debugging.**  Separate from the semantic feedback loop, we also implement a process for handling runtime errors that produce no simulation outputs. If the generated Python code fails to execute due to an error (e.g., from incorrect API usage or unexpected perception tool outputs), we automatically capture the full error traceback. A VLM is then prompted to act as a debugger. It is provided with the original flawed code, the full environment and API specifications, and the captured traceback. Its sole task is to correct the code based on the error message, allowing the system to recover from common programming mistakes and increasing the overall success rate of the generation process.

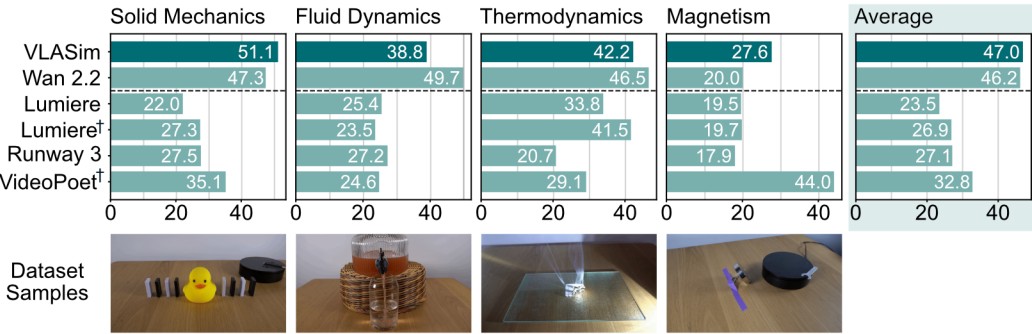

Figure 9: The modified Physics-IQ score of our approach compared with a selection of video generation models. We exclude evaluations on visual-based phenomena, and as such we do not consider the *Optical* category of Physics-IQ, as well as the *MSE* metric for the evaluation. A sample from each category is shown beneath the results figures. † indicates methods which take videos as inputs.

## 4 EXPERIMENTS

We conduct a comprehensive set of experiments to validate our approach. Our evaluation is structured around four primary goals. First, we assess the physical plausibility of our generated simulations and benchmark them against state-of-the-art video models. Second, to demonstrate the versatility of our approach, we showcase results across a wide variety of physical phenomena, such as rigid-body dynamics and fluid interactions. Third, we validate that our method's programmatic output creates an interpretable scene abstraction that users can directly interact with and modify. Finally, in Sec 6.2 we conduct ablation studies to analyze the contribution of each core component of our system.

**Implementation Details.** Our implementation uses Gemini (Comanici et al., 2025) as the core Vision-Language Model agent. All GPU computations are performed on NVIDIA H200 GPUs. The inference time for generating a complete world program from a single input image and prompt is approximately 10 minutes. All code will be made publicly available.

**Baselines.** We benchmark VLASim against several state-of-the-art video generation models. Our primary baseline is Wan2.2 (Wan et al., 2025), as it represents the leading open-source model. We also compare with Lumiere (Bar-Tal et al., 2024) and VideoPoet (Kondratyuk et al., 2023). Finally, we include a select number of examples from Veo3 (Google Deepmind, 2025b)[1].

**Benchmark and Metrics.** For quantitative evaluation, we use the PhysicsIQ benchmark (Motamed et al., 2025). This dataset is composed of real-world videos capturing a diverse set of physical phenomena, categorized into the following areas: solid mechanics, fluid dynamics, magnetism, thermodynamics, and optics. As our method focuses on physical dynamics rather than visual appearance, we exclude the optics category from our evaluation.

We adopt three of the four metrics proposed by PhysicsIQ to evaluate motion and action. We exclude the Mean Squared Error (MSE) metric, as our goal is to model physically plausible motion via abstract simulation rather than achieve photorealistic pixel-level consistency. The metrics we use are: Spatial IoU (evaluating where the action happened), Weighted Spatial IoU (evaluating where and how much action happened), and Spatiotemporal IoU (evaluating when and where the action happened). Following the original benchmark protocol, we combine these three components to produce a final score out of 100, where a higher score indicates a more physically accurate prediction.

To specifically evaluate the logical reasoning capabilities of our method on a deterministic, rule-based system, we introduce a benchmark based on Conway's Game of Life (Conway et al., 1970). We created a test set of 10 distinct initial scenes. For each scene, the task is to generate a simulation program from a single input frame that correctly predicts the evolution of the board over subsequent

---

[1]Due to the significant costs associated with Veo3, a full-scale evaluation was prohibitive for this paper.

steps. The accompanying text caption explicitly identifies the scene as Conway's Game of Life, tasking the model to apply the game's known rules. We evaluate the accuracy of the predicted frames using the F1 score, which is computed by comparing the state of each cell (live or dead) in the predicted grid against the ground truth, treating live cells as the positive class.

**Qualitative Results.** Figure 7 presents a selection of our qualitative results across various challenging scenarios. As shown in the top three rows, `VLASim` successfully generates physically plausible simulations for scenes involving complex solid body dynamics, as well as thermodynamics, as well as fluid interactions. The fourth row demonstrates the model's ability to select an appropriate level of abstraction. For this predominantly planar scene, our method correctly infers that a simpler 2D abstraction and simulation is sufficient, generating a program that is both efficient and accurate. The generated abstractions models the important components of the scene while intentionally discarding distracting, high-frequency visual details and appearance. The goal is not to achieve photorealism, but to focus exclusively on producing a plausible and accurate simulation.

Figure 8 provides a direct qualitative comparison of `VLASim` against the state-of-the-art video models, Wan2.2 (Wan et al., 2025) and Veo3 (Google Deepmind, 2025b). While the baseline models generate visually detailed outputs, they often exhibit common physical inconsistencies. For example, in the top row, both baselines change the number of visible blocks in the scene, and do not correctly model the effect of the gap between the blocks. In contrast, `VLASim` generates simulations where the objects behave as entities governed by consistent physical laws. Blocks collide plausibly, highlighting a fundamental advantage of our approach: the baselines attempt to learn physics implicitly within a high-dimensional pixel space, making them prone to such artifacts. Our method, by generating a program for an explicit physics engine, enforces object permanence and consistent dynamics. Further results are provided in Appendix Sec 6.3 and in Supplementary Materials.

**Quantitative Results.** Figure 9 plots the quantitative results on PhysicsIQ. The scores show that `VLASim` performs on par with Wan2.2, the leading open-source video model. However, these quantitative metrics fail to capture the full picture of physical plausibility. As is evident in our qualitative comparisons (Figure 8) and supplementary video, the outputs from Wan2.2 frequently exhibit non-physical artifacts—such as objects unnaturally merging that demonstrate a lack of a true underlying physics model. This discrepancy suggests that the IoU-based metrics of PhysicsIQ, while useful for tracking general motion, are not sensitive enough to penalize these critical, common-sense violations. Our method, which is governed by an explicit physics engine, avoids such artifacts by design, a crucial advantage not fully reflected in the final score. We note that physical prediction is often non-deterministic. To account for this, for both our method and Wan2.2, we generate three distinct outputs and report the best score. Scores for other models are taken directly from the original PhysicsIQ paper, as their models were not available for our re-evaluation.

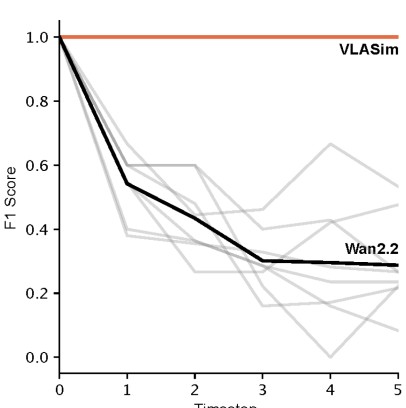

Figure 10: Results on Conway's Game of Life. One timestep corresponds to a single frame step, and an F1 score is calculated between the cells considered 'alive' in the predicted video and in the ground truth.

Finally, to evaluate performance on a purely logical and rule-based task, we present the results of our Conway's Game of Life benchmark in Figure 10. Here, `VLASim` significantly outperforms Wan2.2, achieving a perfect F1 score. This result highlights a fundamental difference between the two approaches. As a pixel-prediction model, Wan2.2 attempts to generate the visual patterns of the game's evolution but consistently fails to adhere to the strict, deterministic rules, leading to cumulative errors. This demonstrates the inherent advantages of an explicit, program-synthesis approach for tasks that require precise, rule-based reasoning.

**Interventions** A central advantage of `VLASim` over video generation methods is its use of an interpretable simulator for prediction, allowing fine-grained intervention through modification of the

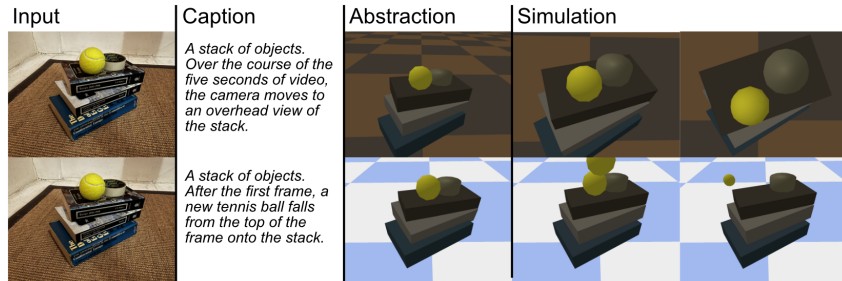

Figure 11: Intervention experiment with a stack of books. By changing the caption of the video passed to `VLASim`, we can change the kinematics of the resulting simulation.

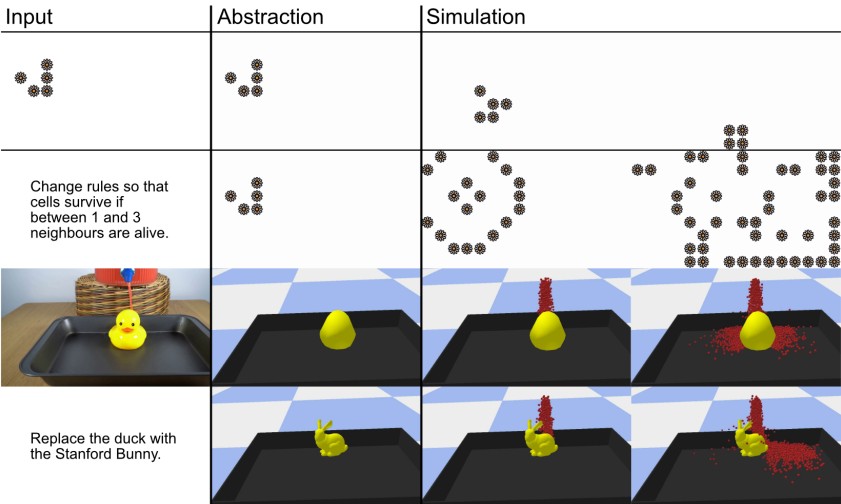

Figure 12: Intervention experiment with fine control. In the first example, we intervene to change the rules of Conway's Game of Life. In the second example, we change duck to the Stanford Bunny.

`Simulation` class, in addition to coarse intervention through captioning which is also available in video generation. Figure 11 shows one example of coarse intervention via text conditioning.

Figure 12 demonstrates examples of fine-grained intervention. In these two scenarios we demonstrate several interventions, made by directly editing the Simulation code. In the first example we show modifications to Conway's Game of Life played with flowers, changing the rules required for cells to survive between timesteps. This demonstrates that we are able to use intervention to generate scenarios which lie outside the training data of the language model, as the modified scenario does not constitute Conway's Game of Life. The second example shows an intervention where a duck figure is replaced in simulation to the Stanford Bunny (Turk & Levoy, 1994), while remaining physically plausible. Further examples of intervention are shown in appendix Section 6.4.

## 5 CONCLUSION

In this work we introduced `VLASim`, a new paradigm for building dynamic world models from static images. We have shown that by tasking a Vision-Language Model with world program synthesis, it is possible to generate explicit, executable simulations that are physically plausible, interactive, and versatile. Our experiments demonstrate that this approach avoids common physical artifacts of pixel-prediction models and excels at tasks requiring precise, rule-based reasoning. This programmatic approach represents a significant step towards creating more grounded and interactive world models. We believe our work points to a broader shift in how we build autonomous agents. Instead of relying on monolithic, end-to-end models that learn an opaque representation of the world, `VLASim` functions as a compositional agent that reasons about the world and writes code to model it.

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

# 6 APPENDIX

## 6.1 QUANTITATIVE RESULTS

In table 1 we provide the numbers that were used in the plots in Figure 9.

Table 1: The modified Physics-IQ score of our approach compared with a selection of video generation models. We exclude evaluations on visual-based phenomena, and as such we do not consider the *Optical* category of Physics-IQ, as well as the *MSE* metric for the evaluation. † indicates methods which take videos as inputs. * Indicates our method

| Physics-IQ | Solid Mechanics | Fluid Dynamics | Thermodynamics | Magnetism | Average |
|---|---|---|---|---|---|
| VLASim* | **51.1** | 38.8 | 42.2 | 27.6 | **47.0** |
| Wan 2.2 | 47.3 | **49.7** | **46.5** | 20.0 | 46.2 |
| Lumiere | 22.0 | 25.4 | 33.8 | 19.5 | 23.5 |
| Lumiere† | 27.3 | 23.5 | 41.5 | 19.7 | 26.9 |
| Runway 3 | 27.5 | 27.2 | 20.7 | 17.9 | 27.1 |
| VideoPoet† | 35.1 | 24.6 | 29.1 | **44.0** | 32.8 |

## 6.2 ABLATIONS

We conduct ablation studies to analyze the contribution of each core component of our system. The results, summarized in Figure 13, demonstrate the importance of each module. We use the same metric as in the main paper, and evaluate on one split of PhysicsIQ dataset. First, we evaluate a variant of our model without access to the perception toolbox API ('No API'). This version performs poorly, both quantitatively and qualitatively, failing to generate coherent or accurate simulations

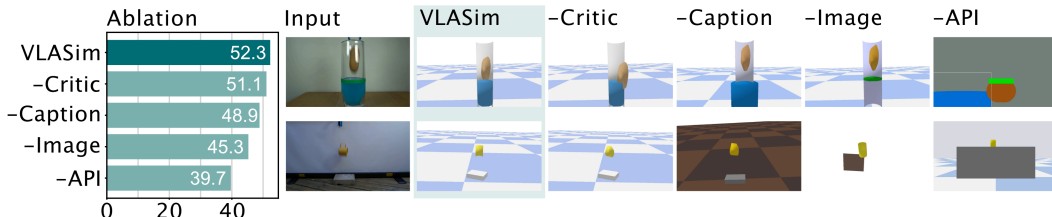

Figure 13: Ablations. We show quantitative results on left, and qualitative on the right. Results demonstrate the utility of each component in our approach.

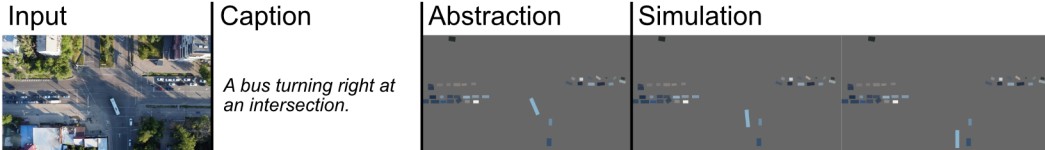

Figure 14: A simulation of overhead traffic with a large number of vehicles. Sample from Aerial Traffic Dataset (SovitOpenCVUniversity, 2025)

which match the input images. This result confirms that the VLM's ability to ground its reasoning in the explicit scene information provided by the perception tools is critical to its success.

Next, we analyze the impact of the critic-and-refinement loop ('No Critic'). Quantitatively, this variant performs similarly to our full method on the PhysicsIQ benchmark. However, we observe a noticeable improvement in the visual quality and physical plausibility of the final simulations when the critic is enabled. This suggests that while the initial program generated by the VLM is often functionally correct, the refinement loop is crucial for correcting subtle errors and improving the overall quality of the simulation.

We additionally ablate the provision of the input image and caption to the generation and critic stages (the image is always available to the API toolbox). These ablations show that both components are used effectively by VLASim, but that visual context is more important for producing an accurate simulation.

### 6.3 FURTHER QUALITATIVE RESULTS

Here we detail further experiments carried out in addition to those described in the main text. In these scenes we focus on interactions which are more challenging to simulate.

**Complexity** We demonstrate several scenes which contain higher complexity of object interaction. In Figure 14 we show a scene involving a large number of vehicles viewed from overhead, along with the direction to simulate the bus turning right at the intersection.

We further show several scenes which are very cluttered to show the generation of multiple objects in the wild in Figure 15. In each case we use the prompt of a tennis ball being thrown into the scene. In an additional example, we show a cluttered desk with a computer mouse on a desk. Here, the model is prompted to wake the computer by moving the mouse. The method shows an understanding of the required complexity of the simulation, only segmenting and modelling the mouse and the computer screen.

**Coupled Dynamics** In Figure 16 we demonstrate two challenging physical processes to simulate. The first example, a double pendulum, is challenging to model as it is a chaotic scene. The second example requires accurately modelling non-inertial reference frame physics. The double pendulum data sample was formed by compositing components onto a background from the Stanford Backgrounds Dataset (Gould et al., 2009). The car image in the second example is taken from the PhysGen Dataset (Liu et al., 2024).

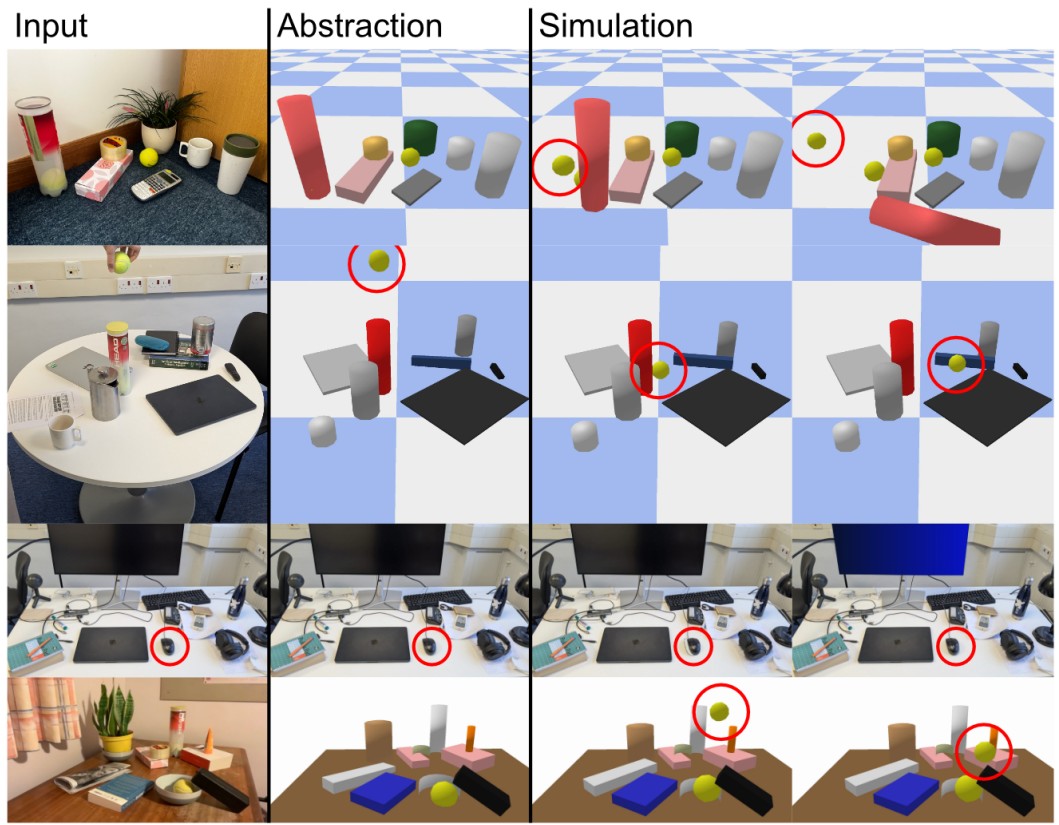

Figure 15: Scenes showing cluttered environments. In three samples the model is prompted to throw a tennis ball into the scene, showing multiple object segmentation and modelling. In the scene with the desktop, the model is prompted to move the mouse to wake the computer. Red circles added for emphasis.

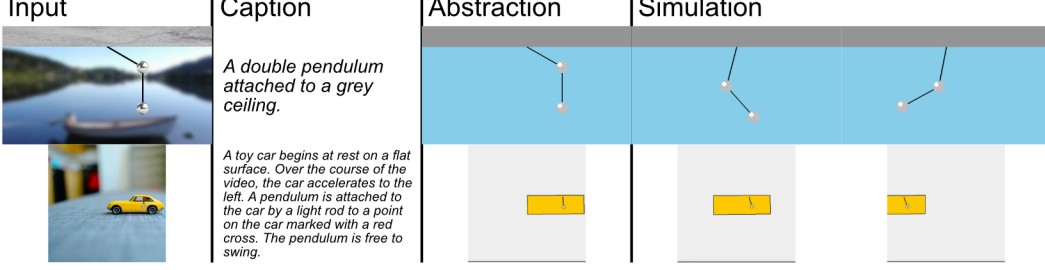

Figure 16: Two examples of physical systems involving coupled dynamics.

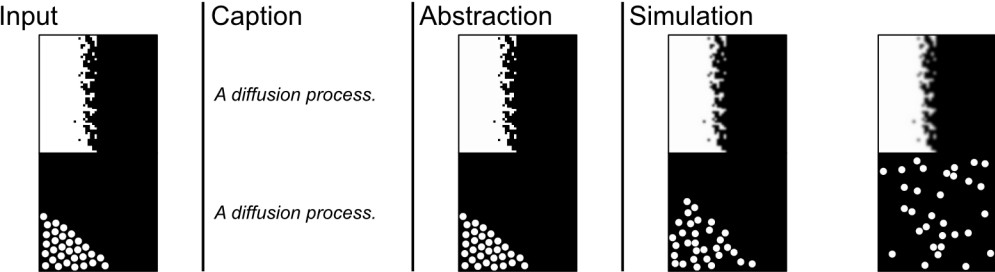

Figure 17: Two different simulations of a diffusion process. In this case, the model must rely on visual context to select the appropriate simulation.

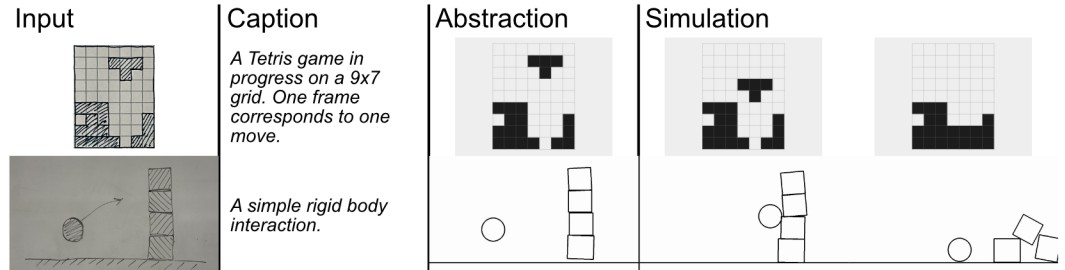

Figure 18: Two examples of physical systems where the supplied image is only an illustration of the target system.

**Visual Context** In Figure 17 we show two example scenes, both of which are supplied with an identical text caption. In this example, the vision model correctly selects the most appropriate interpretation of a diffusion simulation based on the visual context. In the first example the system is modelled as a concentration gradient which flattens over the diffusion process. In the second example the system is modelled with individual particles undergoing Brownian motion, diffusing through a domain.

**Abstraction** Finally, in Figure 18 we demonstrate two scenarios which require `VLASim` to perform the task of abstraction from an illustration of the target system. We use two examples. First, a user prompts our method with a sketch of a game of Tetris. Leveraging its understanding of the rules of the game of Tetris, our approach accurately simulates the future states of the game. In the second, a sketch indicates the process of firing a ball into a stack of cubes. This is accurately modelled by our system.

## 6.4 INTERVENTION EXPERIMENTS

We demonstrate further examples of systems where we have performed an intervention to change the nature of the kinematics, dynamics, or appearance of the simulated scenes.

**Fine-Grained Control** Our interventions can extend to the specific motion of objects or actors in our scenes. This provides a particularly useful application of our method to robotics. In Figures 19 and 20 we show two scenes from the Language Table dataset (Lynch et al., 2022). `VLASim` is able to produce a correct abstraction of the scene which can subsequently be controlled by the user via waypoints. We show two examples of user-provided trajectories for each scene.

**Reaction-Diffusion** A reaction-diffusion system describes how two or more substances interact locally via a chemical reaction while simultaneously spreading out across space via diffusion. Alan Turing theorized these systems could explain the formation of biological features like the spots on a leopard or the ridges on fingerprints in his work "On the Chemical Basis of Morphogenesis" (Turing, 1952). We show how the user can simply intervene in a simulated scene of such a process using our method, adjusting the parameters of the reaction to suit their needs. These are shown in Figure 21.

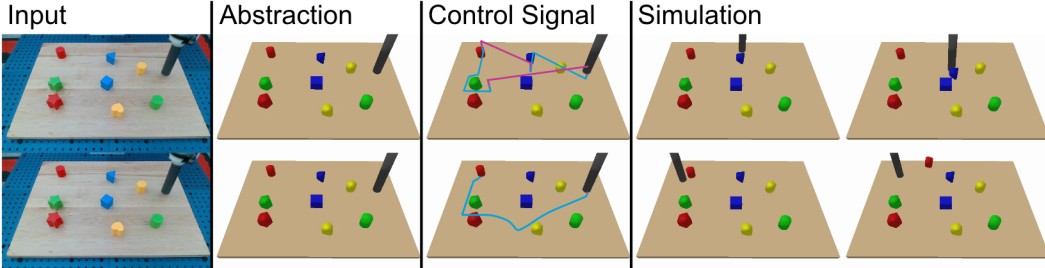

Figure 19: Example of `VLASim` on the Language Table dataset (Lynch et al., 2022). After producing the abstraction, a user can control the trajectory of the robotic mover tool via waypoints.

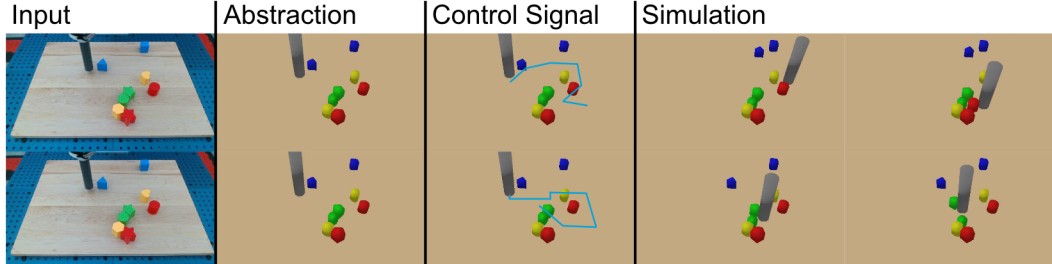

Figure 20: Example of `VLASim` on the Language Table dataset (Lynch et al., 2022). After producing the abstraction, a user can control the trajectory of the robotic mover tool via waypoints.

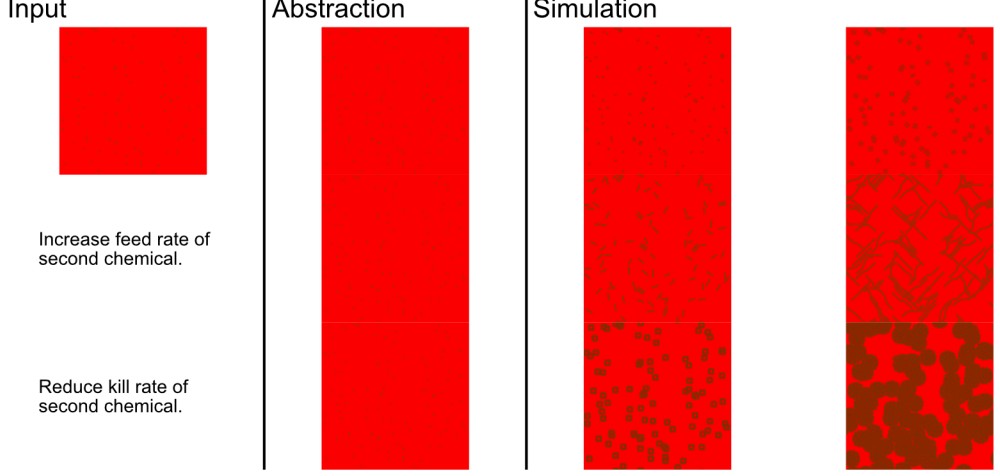

Figure 21: By simply adjusting parameters, the user can change the simulation of a reaction-diffusion process.

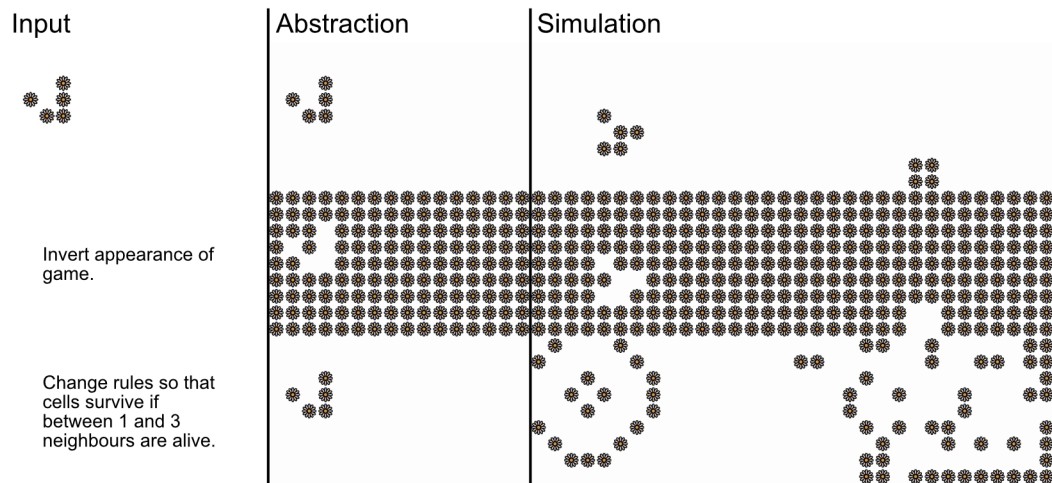

Figure 22: By simple adjustments to the code, the user can change the appearance or rules of Conway's Game of Life.

**Moving Simulations out of Distribution**    In a similar process, the user can intervene to change the appearance or rules of Conway's Game of Life. In Figure 22 we demonstrate the user updating the rules of Conway's Game of Life so that cells which are alive in timestep `t` will survive to step `t+1` if $1 \leq n \leq 3$ cells in its neighbourhood are alive. Importantly, this simulation is no longer Conway's Game of Life, so this intervention represents the user moving the simulation outside of the generation distribution of the language model used to generate the simulation. Additionally, the user can trivially invert the appearance of the Game, so that dead cells appear as flowers, whereas in the input image live cells appeared as flowers.

**Updating Physics**    In further examples, we demonstrate the user intervening in several 3D scenes. In Figure 23, we show an intervention in a scene with a ball colliding with a duck object. Interventions include changing the mass of the duck, changing the direction of gravity, and changing the pose of the camera to a top down view.

In Figure 24 we show several interventions in a scene in which liquid pours onto a plastic duck. Interventions include reducing the flow rate of the liquid, changing the color of the liquid, and changing the duck object to a model of the Stanford Bunny.

In Figure 25 we demonstrate several interventions in a scene with a row of dominos. The interventions are both physical (such as adding and removing blocks), and visual (such as rotating the camera pose).

## 6.5    DISCUSSION OF METRICS

We note that our method performs comparatively with video generation models, even though our abstraction method places it at a disadvantage for the metrics used to calculate Physics-IQ score. In particular, despite the exclusion of MSE as a metric, the use of abstractions, particularly the use of particle-based simulation for fluid flow, means that often simulations produced by `VLASim` do not compare favourably to ground truth, despite the physical simulation being technically accurate. This is illustrated in Figure 26. In this case, the use of particles to represent the cloud of smoke means that our approach is at a disadvantage for metrics which consider changes in pixel value.

Benchmarking for physical plausibility is an active area of research, and we hope that a more appropriate metric will presently be available to evaluate systems such as ours.

## 6.6    INTER-SIMULATION VARIABILITY

It is the nature of modelling physical processes from images that the system which is defined from the image is under-constrained. A model which is capable of truly modelling these systems should

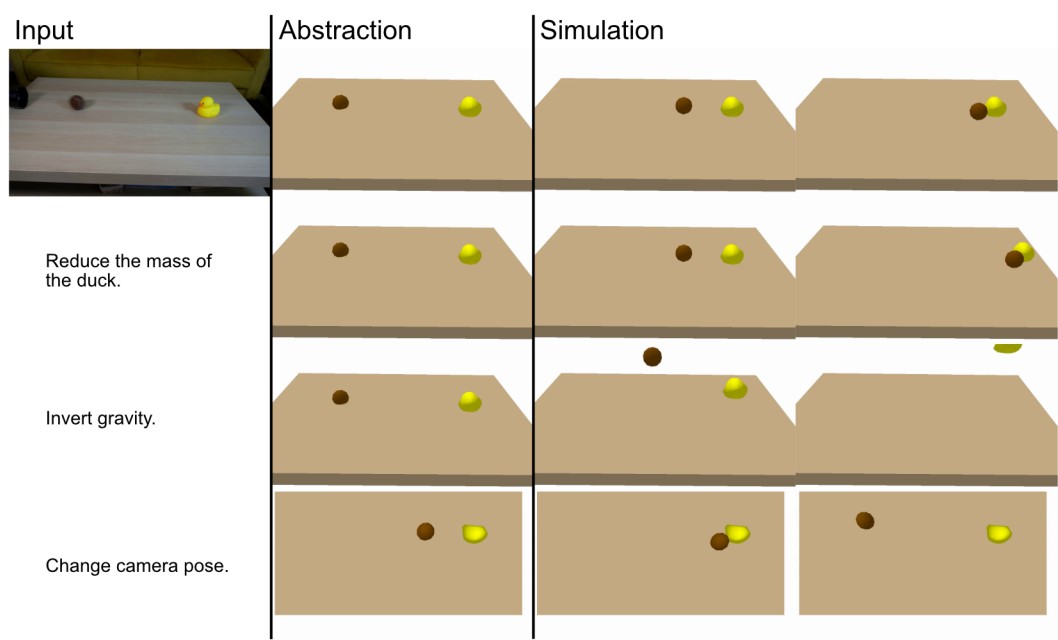

Figure 23: Interventions in a scene in which a ball collides with a duck.

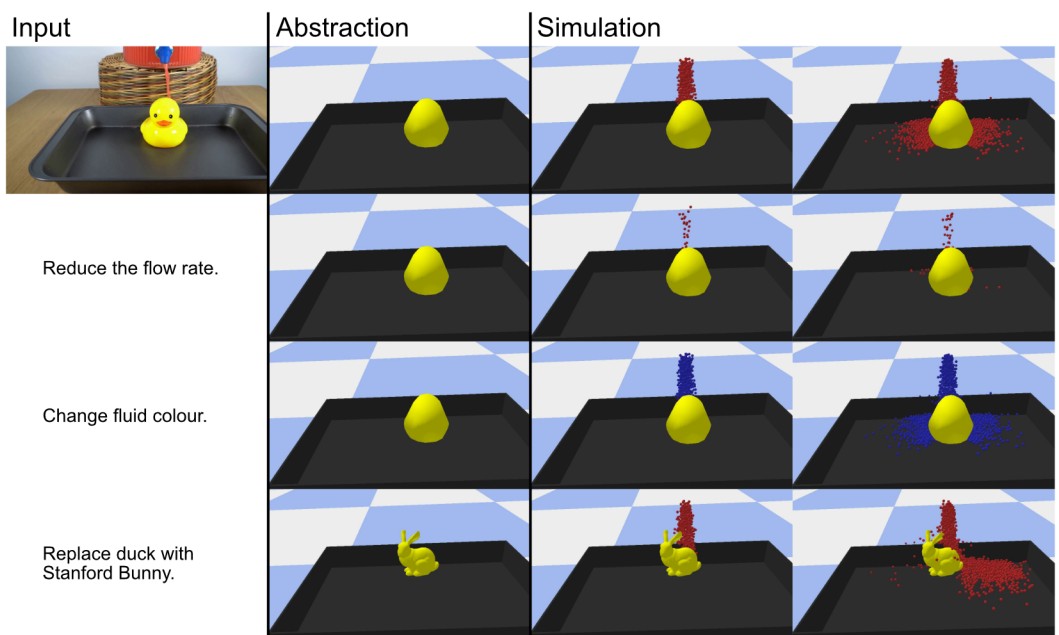

Figure 24: Interventions in a scene in which liquid pours onto a model of a duck.

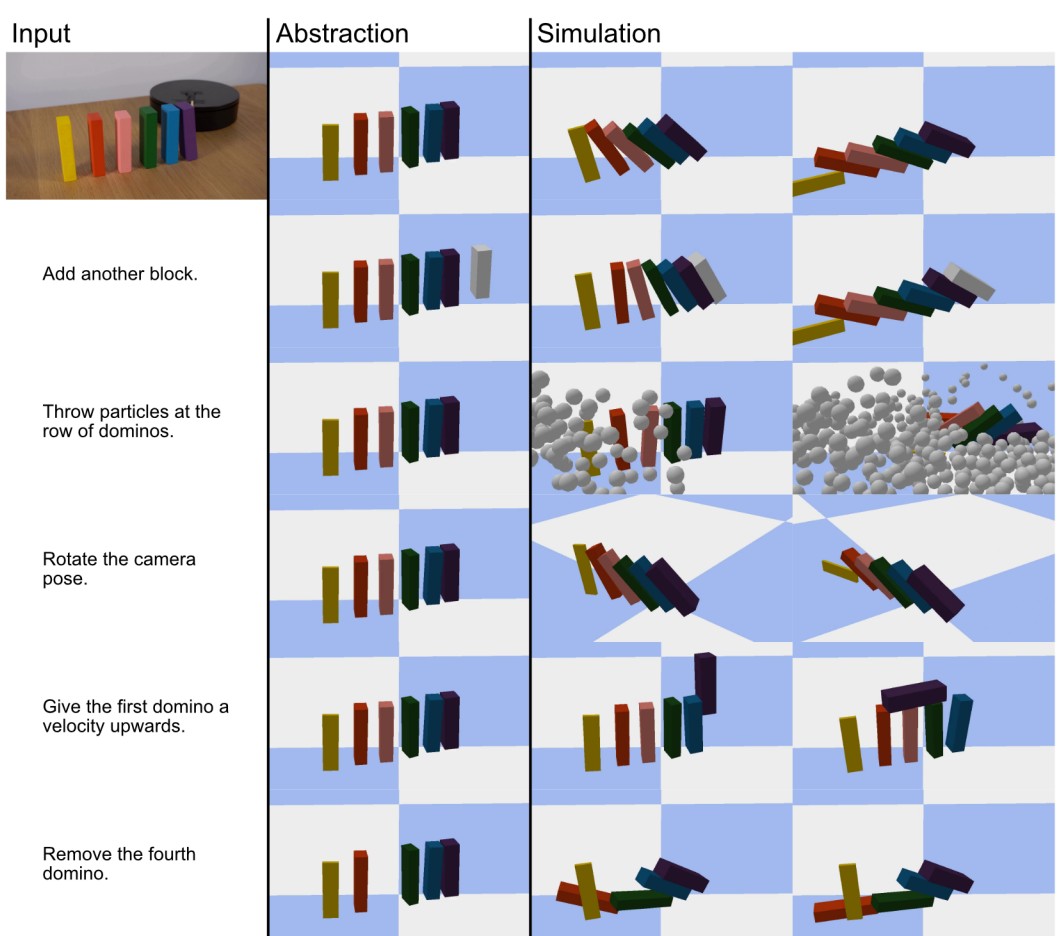

Figure 25: Interventions in a scene consisting of a row of dominos toppling.

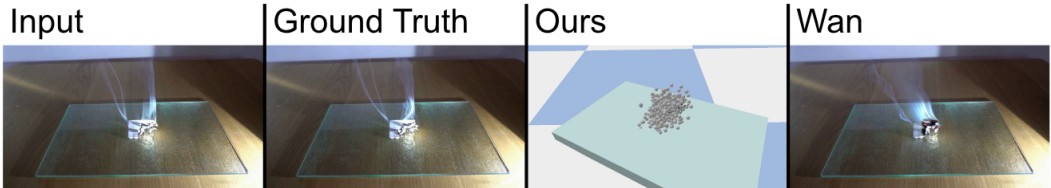

Figure 26: Comparison of `VLASim` with Wan 2.2 for a paper smoke experiment. In this scene, the use of particles to represent the cloud of smoke places abstraction and simulation approaches at a disadvantage for measuring distance to ground truth using pixel-based methods.

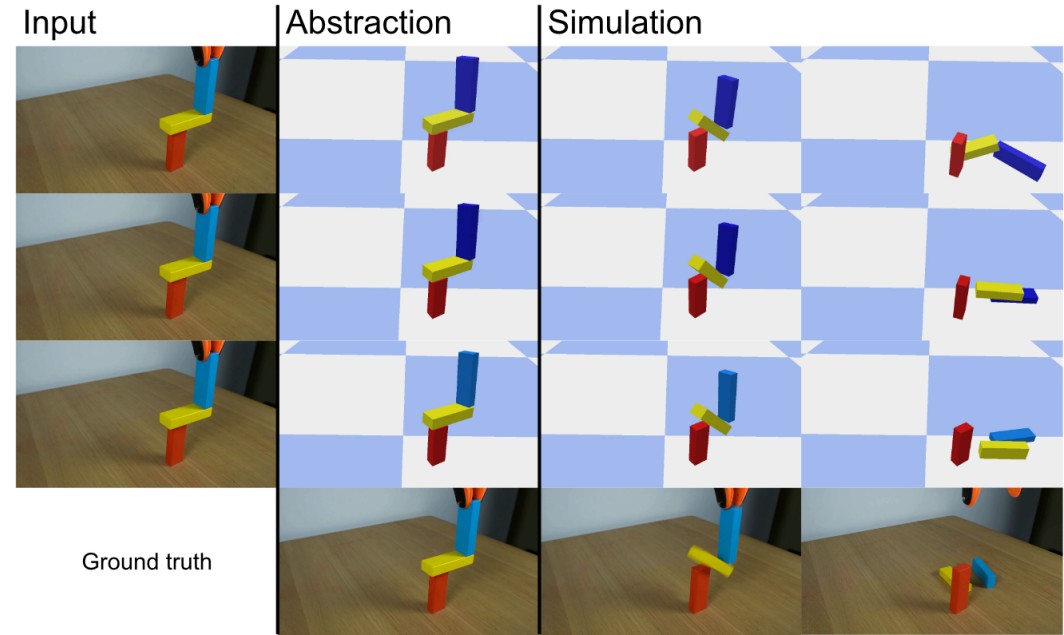

Figure 27: Inter-simulation variability. In repeated simulations of the scene, the core mechanics remain consistent, but the final simulation state predicted changes. Stochasticity in these systems is principally due to the stochastic nature of language modelling.

be able to accurately model this stochasticity. The degree of stochasticity should match the degree of uncertainty dictated by the image of the system. In Figures 27 and 28 we demonstrate that our approach does inherently carry stochasticity. Language models are by construction stochastic, with tokens being selected based on a probabilistic distribution of tokens. This is the principal source of variation between different simulations of the same scene. In the first example, shown in Figure 27 the three different runs of our simulation produce three near-identical outcomes, the core distinction being the final arrangement of blocks due to specifics of falling mechanics. In the second example, shown in Figure 28, there is a higher degree of uncertainty since the initial speed of the ball, coefficients of restitution, and rolling resistance cannot be directly inferred from the images. This leads to a wider distribution of final outcomes, and a larger gap between the simulated outcomes and the ground truth.

## 6.7 LIMITATIONS

A key limitation we observe is the system's sensitivity to errors in the upstream perception toolbox. As a compositional system, the quality of the final simulation is often contingent on the accuracy of tools like segmentation and depth estimation. A failure in one of these modules—for example, misidentifying an object's shape or its 3D position—can lead the VLM to generate a semantically incorrect world program, even if that program is syntactically valid. The VLM currently has no mechanism to question or correct a faulty tool output.

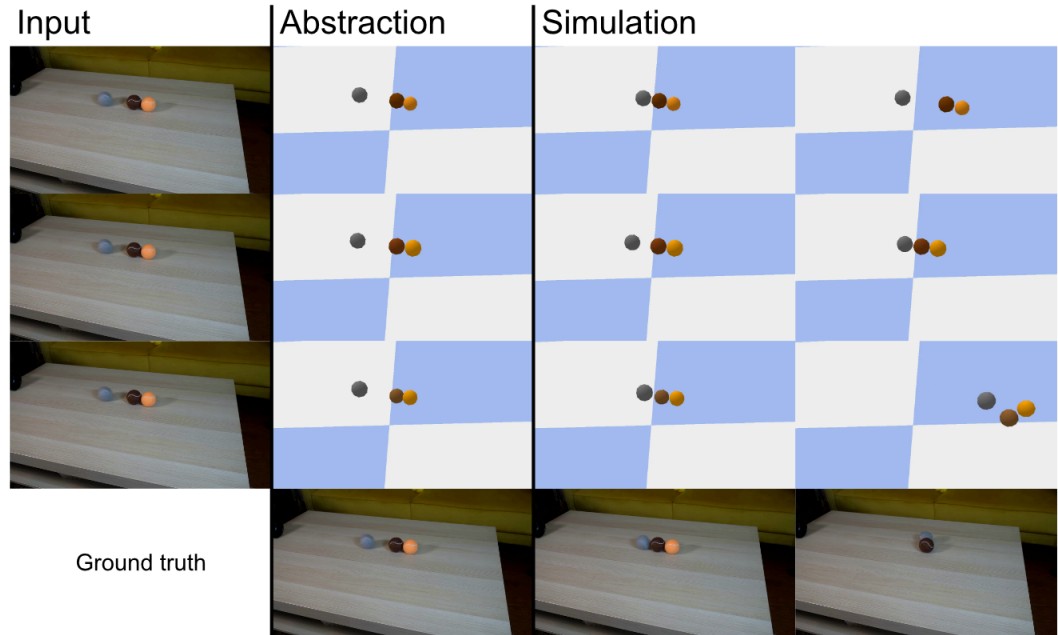

Figure 28: Inter-simulation variability. In repeated simulations of the scene, the final simulation state predicted changes by a greater margin, since physical parameters like ball speed and coefficients of restitution are more difficult to infer from the input image.

