# OpenReview forum: "VLASim: World Modelling via VLM-Directed Abstraction and Simulation from a Single Image"
_ICLR.cc/2026/Conference — Submitted to ICLR 2026_

### Official Review · Reviewer_4CtG · 2025-10-29

**Soundness:** 2
**Presentation:** 2
**Contribution:** 2
**Rating:** 2
**Confidence:** 4

**Summary:**

This paper proposes VLASim, which utilize a VLM as an intelligent agent for distilling a single image into attractable, abstract representation optimized for simulation. It is made of intelligent abstraction, adaptive simulation, and inferred dynamics. Experiments show that VLASim avoids physical artifacts of pixel-prediction models and excels at tasks requiring precise, rule-based reasoning, which provides high-quality simulations across a wider range of dynamic scenarios.

**Strengths:**

1. Paper is easy-to-follow.
2. It is a critical problem of physical interaction in the simulation of world model and build an explicit, structured world representation, in which VLM acts to construct a grounded representation.
3. The idea of combine VLM and real-world simulator is new with explicit representations.

**Weaknesses:**

1. As real scenes are abstracted by simulator, it may be hard to contain all details in the real world. Tiny details may contain influences for real world applications, which may be ignored by the simulator.
2. It seems some foundation models like Gemini Perception, VGGT are utilized for abstraction. This should be illustrated in method overview for clearer expression. Generation Prompt in Figure2 is too general with overlooked details.
3.  It is hard to understand Figure 9. Experiments are too rough and the details of some values cannot be seen. Besides, only comparing with video generation models is not enough considering the mechanism used in your approaches. It seems VLASim is worse than Wan2.2 in Fluid Dynamics and Thermodynamics. More comparisons with simulators are required.
4. Lack ablations to verify the effectiveness.
5. More visualizations and examples are required to show your design.

Minor issue:
Incorrect citation format at line 37-38 in abstract.

**Questions:**

1. Why is VLASim worse than Wan2.2 in Fluid Dynamics and Thermodynamics? Please explain the results.

2. The ablations are missing.

---

> ### Author Response · Authors · 2025-11-22
> **Author Response**
>
> We thank the reviewer for their comments and for noting that the paper is easy to follow, highlights a critical problem, and introduces a new idea combining VLMs with explicit simulators. We address the concerns point-by-point below.
>
> ### (W1) Loss of Fine Details in Abstraction
>
> ---
>
> We do not agree that abstraction causes the system to miss physically important small details. VLASim includes all scene components, large or small, that influence the dynamics. For example, in the balloon–matchstick thermodynamics example, the effect of the thin flame-inducing matchstick is modeled and used to trigger the rupture event. Similarly, in the domino example, the effect of the thin rotating stick is correctly modeled, producing the appropriate forces. These examples illustrate that our abstraction captures all physically relevant structures, regardless of size, while discarding only appearance information that does not affect simulation outcomes.
>
> We do not retain visual appearance details such as texture or color. This is intentional: VLASim produces a simulation-ready abstraction, not a photorealistic reconstruction. Physics engines are designed to operate on simplified geometric primitives, and discarding appearance does not reduce physical fidelity. Instead, abstraction isolates precisely the elements that matter for dynamics.
>
> ### (W2) Overview Figure
>
> ---
>
> We thank the reviewer for this suggestion. In the revised paper, we have expanded **Figure 2** and the method overview to explicitly show the perception modules used (Gemini Perception, VGGT, segmentation, primitive fitting). This should resolve the ambiguity regarding the role of foundation models in abstraction.
>
> ### (W3, Q1) Figure 9 and Comparisons
>
> ---
>
> We appreciate the feedback. We have clarified **Figure 9** by increasing the resolution, exposing the numerical values more clearly, and adding a direct reference to the detailed results table in the appendix. These updates should resolve any ambiguity around the exact numbers.
>
> To the best of our knowledge, there are no existing methods that perform general world modelling by constructing a simulation-ready world directly from a single real image and text description. Classical physics simulators require full state specification and therefore cannot be used as image-to-simulation baselines. For this reason, we follow the established practice on PhysicsIQ and compare against pixel-based video models (e.g., Wan2.2), which are the only available methods that accept visual input and predict future dynamics. If future work introduces directly comparable approaches, we would be very happy to include such comparisons.
>
> For the Fluid Dynamics and Thermodynamics categories, our method uses a particle-based simulation to model these phenomena. This produces physically plausible dynamics, e.g., stable particle trajectories, and consistent interactions with scene geometry, but naturally does not match the exact pixel-level motion of the input video. PhysicsIQ’s IoU-based metrics evaluate similarity in spatial extent, rather than physical correctness. As a result, a pixel-based video model may achieve a slightly higher IoU by visually reproducing the shape or color of the fluid, even when its motion violates basic physics. By contrast, VLASim produces physically valid particle simulations but may diverge in exact visual shape, leading to lower IoU despite higher physical fidelity.
>
> Unfortunately, we are not aware of any existing quantitative metric that can reliably measure physical plausibility of predicted dynamics. Benchmarking physical realism is an active research area, and we expect better evaluation metrics to emerge in the future (see **Section 6.5**).
>
> Importantly, even if one interprets the PhysicsIQ scores as “on par” or slightly lower in some subcategories, VLASim provides significant capabilities that pixel-based models do not:
>
> - **Explicit control:** users can apply arbitrary forces, and modify physical parameters directly in the generated program. These are demonstrated in **Section 6.4**, and the **Intervention Experiments** portion of the Website.
>
> - **Interactivity:** once the world program is synthesized, the environment can be re-simulated instantly under new actions without additional neural inference.
>
> - **Explainability:** all intermediate representations (segmentations, primitives, physics parameters) are explicit and interpretable, enabling inspection, debugging, and modification.
>
> ### (W4, Q2) Missing Ablations
>
> ---
>
> Our original submission already included an ablation study in the appendix (**Section 6.2, Figure 9**). We have now conducted additional ablations, where we study the role played by the image and the text inputs. We find that while both are important for the final quality, the image input plays a more significant role. Please let us know if you think any additional studies are important.
>
> ---
>
> Thanks for pointing out the incorrect citation format. They have been fixed.

---

> ### Author Response · Authors · 2025-11-26
>
> Thank you again for your thoughtful comments. We believe we have addressed all your questions in our response. With the discussion period ending in less than a week, we would be grateful if you could take a look and let us know if there is anything further we can clarify.

---

### Official Review · Reviewer_MkVn · 2025-10-29

**Soundness:** 3
**Presentation:** 3
**Contribution:** 3
**Rating:** 6
**Confidence:** 3

**Summary:**

This paper focuses on the problems of physical logic violations and lack of interactivity in generative video models for world modeling. It also points out the limitations of 3D scene reconstruction methods, which struggle to adapt to abstract 2D environments and cannot support physical simulation. The paper proposes the VLASim framework to address these challenges.

VLASim uses a visual language model as its core intelligent agent. It autonomously selects visual tools to construct abstract representations of 2D or 3D scenes, matches them with compatible physical simulators, and infers potential dynamics from static scenes, achieving structured world modeling.

Experiments show that VLASim performs excellently in the PhysicsIQ benchmark's physical phenomenon simulation and Conway's Game of Life rule reasoning tasks, generating physically reasonable and logically accurate simulation results, outperforming mainstream generative video models.

**Strengths:**

This paper abandons the approach of directly predicting pixels using generative video models, and shifts to a new paradigm of VLM-guided abstraction + adaptive simulation, fundamentally solving the core problems of physical violations and lack of interactivity in pixel-space models. It also overcomes the limitations of 3D reconstruction methods, which cannot adapt to abstract scenes and lack physical simulation capabilities.

VLASim can autonomously adapt to both 2D and 3D scenes. The generated structured programs allow users to modify actions to explore diverse future states. Compared to methods with fixed simulation types, it has a wider range of applicable scenarios and higher interactive value.

**Weaknesses:**

As a composite system, VLASim's simulation quality heavily depends on the output of perception tools such as segmentation and 3D reconstruction. If these tools misidentify object shapes or 3D positions, VLM will generate semantically incorrect world programs, and VLM currently lacks a mechanism to correct these errors.

The inference time for generating a complete world program from a single image and text prompt is approximately 10 minutes, far exceeding the inference speed of mainstream generative video models, making it difficult to meet the application requirements of real-time or low-latency scenarios.

Comparisons with Veo3 only include a limited number of examples; a full-scale evaluation was not conducted due to cost considerations. Furthermore, PhysicsIQ's IoU metrics cannot fully capture physical plausibility (such as non-physical phenomena like object fusion), potentially underestimating VLASim's advantages over baseline models.

Existing experiments primarily focus on single-physical-phenomenon scenarios (such as single rigid bodies or single fluids), failing to demonstrate VLASim's performance in complex multi-object interaction scenarios (such as nested rigid body collisions or multi-fluid mixing), resulting in insufficient verification of its generalization capabilities.

**Questions:**

Regarding simulation issues caused by errors in upstream perception tools, are there plans to design a "tool output verification" mechanism for VLM in the future? For example, could VLM determine the rationality of tool outputs through cross-validation with multiple tools (such as matching segmentation results with 3D point clouds), or adjust the abstraction strategy when tools malfunction?

Is the main bottleneck of the 10-minute inference time the process of VLM generating the world program, or the computational time consumed by perception tools such as 3D reconstruction and segmentation? Are there specific optimization schemes (such as model quantization and tool acceleration) to reduce latency?

VLASim performs excellently in scenarios with clear rules, such as Conway's Game of Life, but how can the model ensure the correctness of simulation logic in scenarios with ambiguous rules (such as non-Newtonian fluid flow)? Is it necessary to introduce domain knowledge or additional rule inputs?

The paper mentions that users can modify actions to explore diverse futures. Are there limitations to the currently supported "custom actions" (such as only supporting the application of simple forces, or being able to adjust physical parameters such as the friction coefficient)? Are there plans to support interaction methods that describe actions in natural language in the future?

---

> ### Author Response · Authors · 2025-11-22
> **Author Response 1/2**
>
> We thank the reviewer for their thoughtful comments. We appreciate the recognition of VLASim’s conceptual shift from pixel prediction to structured program synthesis, its ability to handle both 2D and 3D scenes, and its interactive value. We address the reviewer’s concerns below.
>
> ### (W1) Dependence on Perception Tools
>
> ---
>
> We agree that VLASim relies on the accuracy of its perception toolbox, and we emphasize that this is a feature rather than a limitation. Unlike end-to-end video models where failures are buried inside uninterpretable latent representations, errors in VLASim are explicit and interpretable, because intermediate outputs such as segmentation masks, 3D fits, and geometric primitives can be directly inspected.
>
> VLASim naturally inherits advances in perception tasks. We therefore view reliance on high-quality perception modules as a strength of the approach and a practical path toward increasingly robust world modeling.
>
> ### (W2) Inference Time
>
> ---
>
> The 10-minute generation cost is a one-time program-synthesis step, not per-frame inference. For comparison, high-end video models such as Wan2.2 can require up to ~30 minutes to generate a 5 second video sequence on similar hardware. In contrast, once VLASim has constructed the abstract scene representation, all subsequent simulations run efficiently in a conventional physics engine with no further neural network calls. This yields fast, interactive rollouts for alternative actions, an advantage not available in pixel-based models.
>
> We note that we have not yet optimized our implementation for efficiency, and some of the inefficiency is due to unoptimized implementations of routines like RANSAC. We expect significant speed improvements with a better optimised version.
>
> ### (W3) PhysicsIQ Metric
>
> ---
>
> We agree with the reviewer that the PhysicsIQ IoU-based metrics do not perfectly capture physical plausibility. However, PhysicsIQ is currently the most widely used and best-established benchmark for evaluating physical prediction, and it provides the only standardized quantitative framework available at present.
>
> Benchmarking physical plausibility is an active and open research area, and we expect more suitable metrics to emerge in the future. We have clarified this context in the revision and emphasized the complementary role of our qualitative comparisons, which highlight the physically coherent behavior produced by VLASim that is not fully reflected by existing IoU-based scores (**Section 6.5**).
>
> We provide further results comparing our approach to Veo3 on novel scenes (**Further Comparisons with Veo3 on Website**). We show that Veo3 comprehensively fails to model even simple collision mechanics, whereas our approach is capable of modelling complex kinematics such as double pendulums.
>
> ### (W4) Simple Scenes
>
> ---
>
> While our main paper focuses on controlled scenes from PhysicsIQ in order to isolate physical correctness, VLASim is not limited to these settings. We now include additional real-world examples (cluttered office scene, driving scene, multi-object interactions, multi-fluid mixing such as “juice in water”, **Section 6.3/Fine Grained Control, Further Comparison with Veo3, Further Results, Coarse Control on Website**) in the supplementary material. As this work represents the first step toward vision-conditioned programmatic world models, we focused our quantitative evaluation on the established Physics-IQ benchmark rather than large-scale scene diversity.

---

> > ### Author Response · Authors · 2025-11-22
> > **Author Response 2/2**
> >
> > ### (Q1) Cross-Validation of Tools
> >
> > ---
> >
> > This is a great suggestion! Occasionally, we observe that the critic is able to recognise that an object was not found in the scene, and updates the segmentation prompt to request a different object. Looking at cross-validation between the tools, outside of the critic, would provide useful signal that would be good to explore in future work.
> >
> > ### (Q2) Generation Time Bottlenecks
> >
> > ---
> >
> > The primary bottleneck is querying the LLM, which takes up to 2 minutes per request. Since we run generation, correction, and critique through this model, the time taken can build up. Additionally, routines like RANSAC for primitive fitting take a long time. This is principally because our code is not optimised for speed. We did not explore any quantization and tool acceleration in our work, but we believe they are great avenues for future work.
> >
> > ### (Q3) Dealing With Uncertainties
> >
> > ---
> >
> > For scenarios such as non-Newtonian flow, the toolbox must contain an appropriate model. Our framework can incorporate such domain knowledge by selecting or extending the underlying simulator; the VLM then can then use these extended capabilities (the VLM likely understands the concept of non-Newtonian fluids to select the correct simulator class).
> >
> > The reviewer raises an important point for out-of-distribution rules. When we extend to a scene with out of domain knowledge for the VLM, one would need to provide this knowledge to the VLM. We demonstrate one such example of this in our paper in a new figure. In one scenario we show how the user can intervene to change the rules of Conway’s Game of Life. In the changed rules, a cell survives if 1, 2, or 3 of its neighbours are alive (rather than just 2 or 3). Due to the explicit nature of our simulator this extension to an out-of-domain simulation is possible (**Figure 22/Intervention Experiments on Website**).
> >
> > ### (Q4) Supported Actions for Interactions
> >
> > ---
> >
> > Users can explicitly interact with the simulator by changing any input parameters. We show several new examples in the paper (**Section 6.4/Fine Grained Control, Intervention Experiments on Website**).
> >
> > We have also run experiments where the actions can be specified in text. With this addition, user control can be at a very high-level in the form of text, or very low-level by changing individual parameters of the simulator.

---

> ### Author Response · Authors · 2025-11-26
>
> Thank you again for your thoughtful comments. We believe we have addressed all your questions in our response. With the discussion period ending in less than a week, we would be grateful if you could take a look and let us know if there is anything further we can clarify.

---

> > ### Comment · Reviewer_MkVn · 2025-11-27
> >
> > Thank you to the authors for their detailed replies; I have no further questions.

---

### Official Review · Reviewer_MeXT · 2025-10-31

**Soundness:** 3
**Presentation:** 3
**Contribution:** 3
**Rating:** 6
**Confidence:** 3

**Summary:**

This paper proposes VLASim, a novel world modeling paradigm that shifts from black-box video prediction to explicit, simulation-based program synthesis . VLASim uses a Vision-Language Model (VLM) as an orchestrating agent that, from a single image and text description, generates an executable Python program representing the world . The VLM intelligently selects from a suite of perception tools (e.g., segmentation, 3D reconstruction) to build a tractable 2D or 3D abstract scene representation . It then adaptively chooses a compatible physics engine (e.g., rigid body, fluid, or 2D logic) and writes the simulator code to act upon this representation . The framework includes a VLM-based critic loop for self-correction . Experiments show that this approach produces more physically plausible and logically sound simulations (e.g., in Conway's Game of Life) than state-of-the-art video generation models .

**Strengths:**

1. The paper's core contribution is a significant conceptual shift from implicit pixel prediction to explicit, interpretable world program synthesis . This is a major strength, as the resulting model is structured, queryable, and interactive by design, overcoming limitations of opaque video models.

2. The VLM-driven approach is highly versatile. It can intelligently select the appropriate dimensionality (2D vs. 3D) and the correct physics (e.g., rigid body, fluid, or even abstract logic) for a given scene, rather than being a one-size-fits-all model.

3. The framework truly shines in rule-based, deterministic environments. The experiment on Conway's Game of Life, where VLASim achieves a perfect F1 score by inferring the rules while the SOTA video model fails, is a powerful demonstration of this paradigm's advantage.

**Weaknesses:**

1. The system is critically dependent on the accuracy of its upstream perception toolbox . The paper admits that failures in segmentation or 3D estimation lead to semantically incorrect simulations. The ablation "No API" confirms this is a single point of failure, as the VLM alone cannot ground its reasoning .

2. The claim of modeling from a "single image" is slightly overstated, as the VLM relies heavily on the text prompt (e.g., "The platform rotates clockwise...")  to infer all dynamics. This suggests the VLM is not inferring latent dynamics from the static image itself, but rather translating the text description into a simulation.

3. The examples shown are relatively simple (a few objects) . It is unclear how this agent-based, multi-step synthesis process scales to complex, real-world scenes. The 10-minute inference time to generate a single world program is also high.

4. The paper's main quantitative results on the PhysicsIQ benchmark show VLASim is only "on par" with the open-source baseline Wan 2.2. The authors should qualitatively argue that the benchmark's metrics are flawed and fail to capture the physical plausibility  where VLASim excels . This is a weak empirical position.

**Questions:**

1. If you provide an ambiguous image (e.g., the dominos) with no text description, what latent dynamics, if any, does the VLM infer? Can it predict that the dominos will fall, or is the text prompt doing all the work of describing the dynamics?

2. The paper states the VLM "has no mechanism to question or correct a faulty tool output". Can the critic-refinement loop correct perception errors (e.g., a failed segmentation) or only code errors?

3. The 10-minute inference time is very high. What is the primary bottleneck?

4. How does the VLM "choose" the compatible physics simulator? Is this an explicit reasoning step (e.g., "This scene needs a rigid body solver"), or does it simply write Python code using libraries like PyBullet because they were listed as "Available libraries" in the prompt?

---

> ### Author Response · Authors · 2025-11-22
> **Author Response 1/2**
>
> We thank the reviewer for the thoughtful assessment of our work, especially the recognition of (i) our conceptual shift from pixel prediction to explicit program synthesis, (ii) the versatility of our adaptive 2D/3D simulation framework, and (iii) the strong performance on rule-based systems such as Conway’s Game of Life. We address the reviewer’s concerns below.
>
> ### (W1) Dependence on Perception Tools
>
> ---
>
> We agree that VLASim relies on the accuracy of its perception toolbox, and we emphasize that this is a feature rather than a limitation. Unlike end-to-end video models where failures are buried inside uninterpretable latent representations, errors in VLASim are explicit and interpretable, because intermediate outputs such as segmentation masks, 3D fits, and geometric primitives can be directly inspected.
>
> The perception toolbox is an essential component of the method. As shown in the “No API” ablation (**Figure 13**), removing these tools dramatically degrades performance. The VLM alone is not expected to infer a full physical world program from raw pixels. Our framework explicitly addresses this by grounding VLM reasoning in a mature and explainable suite of vision tools. Since these tools continue to improve rapidly (e.g., segmentation, depth, 3D estimation), VLASim naturally inherits these advances without architectural modification. We therefore view reliance on high-quality perception modules as a strength of the approach and a practical path toward increasingly robust world modeling.
>
> ### (W2) “Single-image” Claim
>
> ---
>
> The main text of our paper correctly states that VLASim conditions on both the input image and its accompanying text caption. Our original title and abstract could have implied otherwise, and we have updated it to avoid this misunderstanding. We appreciate the reviewer for bringing this to our attention.
>
> To clarify the contribution of each modality, we performed additional ablations in which we remove either the image or the text input (**Figure 13**). Both modalities are important for final performance, but removing the image input leads to a substantially larger degradation. This shows that the VLM does infer meaningful latent structure and object relationships from the image. Both modalities play a role in the final performance of our method. This is further demonstrated in our experiments on visual context, in which we provide the same caption to two different illustrations of diffusion. The model correctly uses visual context to infer the correct simulation type for each image (**Figure 16/Using Visual Context on Website**).
>
> ### (W3) Scene Complexity and Inference Time
>
> ---
>
> While our main paper focuses on controlled scenes in order to isolate physical correctness, VLASim is not limited to these settings. We now include additional real-world examples (cluttered office scene, driving scene, and multi-object interactions, **Figures 14, 15, 17, 18, 19, 20, 21/Fine Grained Control, Further Comparisons with Veo3, Further Results, Coarse Control on Website**) in the supplementary material to illustrate that the abstraction–simulation pipeline generalizes beyond tabletop scenarios. As this work represents the first step toward vision-conditioned programmatic world models, we focused our quantitative evaluation on the established Physics-IQ benchmark rather than large-scale scene diversity.
>
> Regarding inference time, the 10-minute generation cost is a one-time program-synthesis step, not per-frame inference. For comparison, high-end video models such as Wan2.2 can require up to ~30 minutes to generate a video sequence on similar hardware. In contrast, once VLASim has constructed the abstract scene representation, all subsequent simulations run efficiently in a conventional physics engine with no further neural network calls at arbitrary resolution. This yields fast, interactive rollouts for alternative actions, an advantage not available in pixel-based models.
>
> Finally, we note that we have not yet optimized our implementation for efficiency, and some of the inefficiency is due to unoptimized implementations of routines like RANSAC. We expect significant speed improvements with a better optimised version.

---

> > ### Author Response · Authors · 2025-11-22
> > **Author Response 2/2**
> >
> > ### (W4) Metrics
> >
> > ---
> >
> > These metrics primarily measure spatial and temporal overlap with the ground-truth video, and do not directly quantify physical plausibility. For example, in the case of fluids or thermodynamics, our method models the scene with particle simulations. Even though the results are more physically plausible, the motion of the particles would not completely agree with the motion of gas or fluid. As the reviewer suggests, we now clarify that benchmarking physical plausibility is an open research problem, and PhysicsIQ provides the best available, but imperfect, indicator (**Section 6.5**).
> >
> > We also emphasize that even if one interprets the PhysicsIQ scores as “on par,” VLASim offers substantial capabilities not present in video models:
> >
> > - **Explicit control and interactivity** (arbitrary forces and parameter edits) enabled by executable programs,
> >
> > - **Explainability** through fully interpretable intermediate representations and physics parameters,
> >
> > - **Consistency** of physical laws enforced by the simulator.
> >
> > These features address core limitations of pixel-based world models and are central to the motivation of VLASim. We have clarified these points more explicitly in the revised paper and supplement (**Section 6.4/Intervention Experiments on Website**).
> >
> > ### (Q1) Dealing with Ambiguities
> >
> > ---
> >
> > We have conducted a new ablation (**Figure 13**) which shows that in fact the image is more useful to the model than the text caption for generation. The video-language model can directly infer the mechanics from an (ambiguous) image based, make assumptions about how the scene might evolve, and we add some results to the supplementary material and the updated ablation chart which highlights this fact.
> >
> > ### (Q2) Correcting Perception Errors
> >
> > ---
> >
> > The critic is not directly exposed to the output of the API toolkit. It only observes the final simulation. Occasionally, we observe that the critic is able to recognise that an object was not found in the scene, and updates the segmentation prompt to request a different object, but this is a more indirect application of what the reviewer suggests in their question. Better design of the critic to help it directly correct the output of the toolkit would be an interesting direction to explore.
> >
> > ### (Q3) Inference Time
> >
> > ---
> >
> > The primary bottleneck is querying the LLM, which takes up to 2 minutes per request. Since we run generation, correction, and critique through this model, the time taken can build up. Additionally, routines like RANSAC for primitive fitting take a long time. This is principally because our code is not optimised for speed.
> >
> > ### (Q4) Selection of the Simulator
> >
> > ---
> >
> > The VLM is directed to choose the most appropriate simulator for the scene (without specifying what that is). Therefore, it leverages heuristics, and presumably makes its choice based on the suitability of the simulator for the mechanics which the VLM believes are important. It makes its choice directly in the code, rather than in text. We direct the reviewer to the generation prompt, which we add to the supplementary material for consideration.

---

> ### Comment · Reviewer_MeXT · 2025-11-26
>
> Thank the authors for their detailed response and the corresponding modifications to the manuscript. I particularly appreciate the new ablation study clarifying the importance of the visual modality, as well as the distinction regarding inference time (one-time synthesis vs. real-time simulation). These explanations have addressed my main concerns, and I will maintain my positive score.

---

> > ### Author Response · Authors · 2025-11-26
> >
> > Thank you for this positive feedback, and we are pleased to have addressed your main concerns. Are there any further modifications we could provide which would convince you to improve your rating of our work?

---

### Official Review · Reviewer_nHoT · 2025-11-01

**Soundness:** 2
**Presentation:** 3
**Contribution:** 2
**Rating:** 4
**Confidence:** 4

**Summary:**

The paper introduces a new pipeline, named VLASim that leverages VLM for world modeling. By autonomously selecting multiple vision tools, the VLM agent constructs the scenes with explicit 2D/3D representations and generates Python program for world simulation. Thanks to the tractable abstract representation, the proposed pipeline can simulate more plausible futures with user-defined actions. It is benchmarked against some existing video generation models on PhysicsIQ and Conway's Game of Life.

**Strengths:**

S1) The paper presents a brand-new pipeline, which is novel compared to the common practice for world modeling. I believe its technique will provide some inspirations to the community.

S2) The effectiveness and robustness of the proposed framework are demonstrated in several scenarios.

S3) The writing is easy to follow.

**Weaknesses:**

W1) The paper argues that previous works are typically limited to simple transformations. However, existing world models for decision making (e.g., DriveDreamer and IRASim) do provide fine-grained action controls. On the other hand, the simulation pipeline in this paper doesn't show its richness in action controls and seems to be still limited in transformations.

W2) My main concern is the abstract representation may lose scalability in representing physics and other real-world properties. The proposed pipeline completely discard texture and background information and exclude visual metrics in its experiments. However, I do believe a scalable general and comprehensive representation is necessary for scalable real-world applications.

W3) Apparently, there is a huge sim2real gap in the simulation results. Also, from the supplementary, the supported scenarios are relatively simple and mostly static tabletop cases.

W4) The highly modular pipeline may also introduce accumulated errors compared to end-to-end models, which is also one of my concerns.

**Questions:**

Q1) The idea of using VLMs and programs for world simulation reminds me of the Genesis project [1] and its related works based on physics engines. Can you discuss the difference?

---

[1] https://genesis-embodied-ai.github.io/

---

> ### Author Response · Authors · 2025-11-22
> **Author Response 1/2**
>
> We thank the reviewer for their feedback, noting our work as a "brand-new, novel pipeline" that will "provide inspirations to the community" and appreciating its effectiveness, robustness, and writing quality.
>
> We address each concern below.
>
> ### (W1) Fine-Grained Control
>
> ---
>
> We respectfully clarify that action control in VLASim is fundamentally different in scope and expressivity from DriveDreamer and IRASim.
> DriveDreamer is restricted to driving-specific action spaces, whereas VLASim is designed for general-purpose world modeling and supports a much broader set of physical phenomena, including rigid-body interactions, fluids, thermodynamics, and 2D logic systems.
>
> IRASim conditions on a known motion trajectory; it does not infer forces or simulate novel, user-specified physical interventions.
> In contrast, VLASim estimates latent motion via physics simulation (e.g., the inferred motion of dominos under forces, **Fig 8**), and, crucially, because our output is an executable program, users may apply arbitrary forces, torques, parameter edits, and interaction events not present in the input (**Section 6.4/Intervention Experiments on Website**).
> This enables a level of interactivity and control that, to our knowledge, is not supported by existing world models.
> In fact we are able to treat IRASim conditioning as a subset of the breadth of events we can support, and show novel results recreating IRASim trajectories on the Language Table dataset (**Figures 20, 21/Fine Grained Control on Website**).
> This demonstrates that our model can also achieve fine-grained trajectory conditioning.
>
> We have expanded the discussion of these works in the revised paper and added further demonstrations of rich action control, including coarse text-driven interventions (**Figure 11/Coarse Control on Website**) and fine-grained transformations via direct simulation edits (Conway’s GoL variations, ball–duck, liquid–duck interactions, domino chain reactions, reaction–diffusion, **Section 6.4/Intervention Experiments on Website, Fine Grained Control on Website**).
>
> ### (W2) Scalability
>
> ---
>
> We would like to clarify that abstraction in VLASim does not limit scalability.
> Physics engines and simulation systems operate on simplified geometric representations; removing textures and backgrounds isolates the physically relevant components and makes simulation more stable and more efficient, especially in cluttered scenes.
> Our results already span a broad set of phenomena—rigid-body dynamics, thermodynamics, fluids, 2D logic, and multi-object interactions.
> To further address the reviewer’s concern, we have added more complex, highly cluttered multi-object scenes (**Section 6.3, 6.4/Further Results, Further Comparisons with Veo3, Fine Grained Control on Website**), even reconstructing scenes from whiteboard drawings, to the supplementary material. Across these diverse settings, VLASim consistently performs plausible simulations.
>
> Finally, photorealistic appearance is not the goal of VLASim.
> Our contribution is simulation-ready abstraction. Appearance can be added (e.g., via video diffusion conditioned on our simulations) without changing the core method.
>
> ### (W3) Sim2real gap and Static Tabletop Scenes
>
> ---
>
> The appearance gap between the input image and the simulated output is intentional.
> VLASim produces a structured, simulation-ready world model, not a photorealistic reconstruction.
> Textures are omitted because they are irrelevant to the underlying physical reasoning.
>
> While video models can generate visually rich outputs, this does not translate to physical fidelity: even in visually simple scenes, they frequently violate object permanence, causality, collision behavior, or deterministic rules (as shown in our PhysicsIQ and Game-of-Life evaluations, **Figure 8/Comparisons with Veo and Wan on Website**). VLASim focuses on dynamics, not appearance complexity, and abstraction is essential for physically coherent behavior.
> As noted above, rendering our simulations back into photorealistic video is an interesting and orthogonal direction for future work.
> Our results already span a broad set of dynamic phenomena and are not limited to static scenes.
> We show rigid-body dynamics, thermodynamics, fluids, 2D logic, and multi-object interactions. To further address the reviewer’s concern, we have added more complex, highly cluttered multi-object scenes, coupled physical systems, and even reconstructing scenes from whiteboard drawings (**Figures 17, 18/Further Comparisons with Veo3 on Website**).
>
> ### (W4) Error Accumulation
>
> ---
>
> We agree that a modular pipeline can introduce errors. However, end-to-end video models also produce significant errors, as seen in our comparisons where they violate simple physical rules. The key difference is that errors in VLASim are interpretable. Each module produces explicit intermediate outputs that can be inspected and corrected. Thus, modularity enables transparency and correction not possible with other models.

---

> > ### Author Response · Authors · 2025-11-22
> > **Author Response 2/2**
> >
> > ### (Q1) Comparison to Genesis
> >
> > ---
> >
> > Genesis differs fundamentally from VLASim in both its goals and its input-output behavior.
> > Genesis provides a high-performance physics engine and a scripting interface for composing scenes and running physical simulations.
> > To the best of our knowledge, its generative pipeline (which has not been released, either as paper or code) focuses on composing assets within a simulator, rather than grounding a world model directly from a real image.

---

> ### Author Response · Authors · 2025-11-26
>
> Thank you again for your thoughtful comments. We believe we have addressed all your questions in our response. With the discussion period ending in less than a week, we would be grateful if you could take a look and let us know if there is anything further we can clarify.

---

### Author Response · Authors · 2025-11-22
**Summary of Additional Results**

We thank all reviewers for their thoughtful and constructive feedback. We are particularly glad that reviewers appreciated the novelty of our approach and its utility, e.g., appreciating that it is a “brand-new pipeline” (nHoT),  “a significant conceptual shift” (MeXT), tackles “a critical problem of physical interaction” (4CtG), that our method is “fundamentally solving the core problems of physical violations” (MkVn). Additionally, reviewers find our writing to be “easy to follow” (nHoT,4CtG).

The reviewers suggested several new experiments and clarifications. We are happy to incorporate all suggestions and provide many additional results in our revision. A summary of these additions is provided below, with references to the corresponding results included in the individual reviewer responses.


## Fine-grained manipulation
---

These scenes demonstrate the degree of intervention possible with VLASim. In these scenes from the Language Table dataset, we show trajectory control of the robotic mover tool in its interaction with the objects on the table.

- Robotic scene 1: **Fine Grained Control on Website, Figure 19 in paper**,

- Robotic scene 2: **Fine Grained Control on Website, Figure 20 in paper**


## Novel Complex Scenes
---

These cluttered scenes demonstrate that our approach is able to handle environments with many different cluttered objects, which can be handled by VLASim.

- Dropping a ball onto a table: **Further Results on Website, Figure 15 in paper**

- Throw ball at arrangement of objects on floor: **Further Comparisons with Veo3 on Website, Figure 15 in paper**

- Move mouse to wake computer on a cluttered desk: **Further Results on Website, Figure 15 in paper**

- Throw a ball into objects on a table: **Further Results on Website, Figure 15 in paper**

Aerial Traffic example demonstrates versatility of method to direct application in driving environment.

- Bus turning right at intersection: **Further Results on Website, Figure 14 in paper**

Diffusion examples demonstrate that our approach is able to use visual context to select the most appropriate simulation, using an image of a concentration gradient in one scene and an image of a Brownian motion scene in another with the same text prompt.

- Diffusion of concentration gradient: **Use of Visual Context in Website, Figure 16 in paper**

- Diffusion of particles in Brownian Motion: **Use of Visual Context in Website, Figure 16 in paper**

Coupled Dynamics show complex physical systems including a chaotic system (the double pendulum) and an image which requires modelling non-inertial reference frame physics (pendulum on car).

- Pendulum on car: **Further Results in Website, Figure 17 in paper**

- Double pendulum: **Further Comparisons with Veo in Website, Figure 17 in paper**

Abstraction experiments require VLASim to understand the intended system from an abstracted, symbolic representation of the system. In this case both systems are drawn on a piece of graph paper and a whiteboard respectively.

- Tetris: **Further Results in Website, Figure 18 in paper**
- Collision on whiteboard: **Further Comparisons with Veo3 in Website, Figure 18 in paper**


## Intervention Experiments
---

Our chemical intervention experiment shows how we can simply change parameters of a chemical reaction process to change a simulation outcome.

- Reaction diffusion: **Intervention Experiments in Website, Figure 21 in paper**

Interventions in PhysicsIQ show how the user can intervene to change a diversity of physical parameters of a simulation. These include replacing objects, changing gravity, and adding additional collision objects.

- Ball hits duck: **Intervention Experiments in Website, Figure 23 in paper**

- Liquid on duck: **Intervention Experiments in Website, Figure 24 in paper**

- Block domino:  **Intervention Experiments in Website, Figure 25 in paper**

Coarse (text-based) Conditioning

- Book stack: **Coarse Control in Website, Figure 11 in paper**

We also show how the user can intervene to move a simulation out of distribution of the LLM by intervening directly in the Simulator.

- Conway’s Game of Life with Flowers: **Intervention Experiments in Website, Figure 12 in paper**


Further Changes
---
- Complete generation and critic prompts have been added to the supplementary material.
- Ablations have been updated to include ablations of one of the image and text components from the generation and critic prompts. These highlight the importance of both components of the prompt to the quality of our overall method.

---

### Meta-Review · Area_Chair_8Qd1 · 2025-12-31

**Summary:**

The paper initially received two positive and two negative reviews. Overall, I believe the authors have addressed most of the raised concerns and questions in a reasonable manner. However, a central issue, namely the gap between physics simulators and the real world, was consistently raised across multiple reviews, including by those with an overall positive assessment.

I find the authors’ arguments generally reasonable. At the same time, I share the reviewers’ concern that the proposed approach remains fundamentally constrained by the capabilities of current physics simulators.

I acknowledge that achieving complete realism is not the primary goal of this work, and I agree that the proposed pipeline is an interesting approach and has promise for incorporating physical priors into the world models. However, given the overall tone of the reviews, I do not believe the current revision and rebuttal are sufficient to change the reviewers’ assessments. Strengthening the discussion or providing more concrete evidence demonstrating how the proposed approach can bridge this gap would make the paper significantly more convincing.

As an area chair, I find these concerns partially convincing, and I also note that even the positive reviewers do not strongly champion this paper. Given the current state of the work and the overall reviewer assessments, I am unable to recommend acceptance at this time. I encourage the authors to further refine the paper and consider resubmission to a future venue.

**Reviewer Concerns:**

See the Summary section.

**Reviewer Scores:**

See the Summary section.

---

### Decision · Program_Chairs · 2026-01-26

Reject